# BEYOND THE BOUNDARIES OF PROXIMAL POLICY OPTIMIZATION

## ABSTRACT

Proximal policy optimization (PPO) is a widely-used algorithm for on-policy reinforcement learning. This work offers an alternative perspective of PPO, in which it is decomposed into the inner-loop *estimation* of update vectors, and the outer-loop *application* of updates using gradient ascent with unity learning rate. Using this insight we propose *outer proximal policy optimization* (outer-PPO); a framework wherein these update vectors are applied using an arbitrary gradient-based optimizer. The decoupling of update estimation and update application enabled by outer-PPO highlights several implicit design choices in PPO that we challenge through empirical investigation. In particular we consider non-unity learning rates and momentum applied to the outer loop, and a momentum-bias applied to the inner estimation loop. Methods are evaluated against an aggressively tuned PPO baseline on Brax, Jumanji and MinAtar environments; non-unity learning rates and momentum both achieve statistically significant improvement on Brax and Jumanji, given the same hyperparameter tuning budget.

## 1 INTRODUCTION

Proximal policy optimization (PPO) (Schulman et al., 2017b) is ubiquitous within modern reinforcement learning (RL), having found success in domains such as robotics (Andrychowicz et al., 2020b), gameplay (Berner et al., 2019), and research applications (Mirhoseini et al., 2021). Given its ubiquity, significant research effort has explored the theoretical (Hsu et al., 2020; Kuba et al., 2022) and empirical (Engstrom et al., 2020; Andrychowicz et al., 2020a) properties of PPO.

PPO is an on-policy algorithm; at each iteration it collects a dataset using the current (behavior) policy. This dataset is used to construct a surrogate to the true objective, enabling gradient-based optimization while seeking to prevent large changes in policy between iterations, similar to trust region policy optimization (Schulman et al., 2017a). The solution to the surrogate objective is then taken as the *behavior parameters* for the following iteration, defining the behavior policy with which to collect the following dataset. The behavior policies are therefore *exactly coupled* with the preceding surrogate objective solution.

In this work we instead consider the inner-loop optimization of each surrogate objective to estimate an update vector, which we name the *outer gradient*. A trivial result follows that the outer loop of PPO can be viewed to update the behavior parameters using unity learning rate $\sigma = 1$ gradient ascent on the outer gradients. Using this insight we propose outer-PPO, a novel variation of PPO that employs an arbitrary gradient-based optimizer in the outer loop of PPO. Outer-PPO *decouples* the estimation and application of updates in way not possible in standard PPO. An illustration of outer-PPO applying a learning rate greater than unity is provided in figure 1. The new behaviors enabled by outer-PPO raise several questions related to implicit design choices of PPO:

**Question 1.** *Is the unity learning rate always optimal?*

**Question 2.** *Is the independence (lack of prior trajectory information e.g momentum) of each outer update step always optimal?*

**Question 3.** *Is initializing the inner loop surrogate objective optimization at the behavior parameters (without exploiting prior trajectory / momentum) always optimal?*

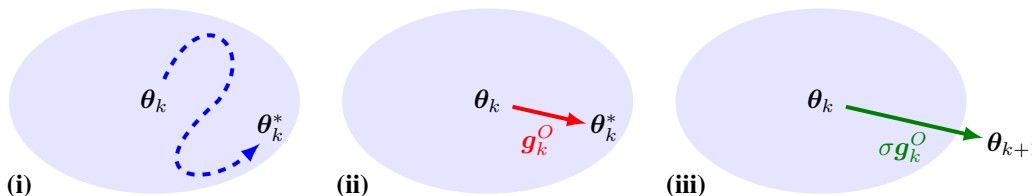

Figure 1: **Diagram of outer-PPO estimating and applying the outer gradient as an update.** (i) Transitions are collected with policy $\pi(\boldsymbol{\theta}_k)$ defining a surrogate objective and corresponding 'trust-region' (shaded) surrounding $\boldsymbol{\theta}_k$; inner-loop optimization of the surrogate objective (blue dashed) yields $\boldsymbol{\theta}_k^*$. (ii) Outer-PPO computes outer gradient as $\boldsymbol{g}_k^O \leftarrow \boldsymbol{\theta}_k^* - \boldsymbol{\theta}_k$. (iii) Outer-PPO updates behavior parameters using an arbitrary gradient based optimizer applied to the outer gradient to give $\boldsymbol{\theta}_{k+1}$, in this case gradient ascent with a learning rate $\sigma > 1$. Standard PPO can be understood as directly taking $\boldsymbol{\theta}_{k+1} \leftarrow \boldsymbol{\theta}_k^*$, or as a special case of outer-PPO corresponding to gradient ascent with learning rate $\sigma = 1$.

This work forms an empirical investigation of the aforementioned questions. To motivate this investigation, consider the clipping parameter $\epsilon$ of PPO, controlling the size of the 'trust region' within which we seek to restrict our update. If $\epsilon$ is set too low, we restrict ourselves to small policy updates. Conversely, if we set $\epsilon$ too high we decrease the reliability of our update direction. In outer-PPO, introducing an outer learning rate decouples these two effects; we are able to reliably estimate an update vector using a moderate $\epsilon$, but then take step of large magnitude in this direction.

We emphasize that we do not seek to identify the most performant configuration possible but to understand the performance of outer-PPO relative to a well-tuned PPO baseline. To this end we restrict the tuning of outer-PPO to simple grid searches applied to fixed base PPO hyperparameters.

Our contributions are as follows:

- We propose *outer proximal policy optimization* (outer-PPO), in which an arbitrary gradient-based optimizer is applied to the 'outer gradients' of PPO. By tracking the outer trajectory, outer-PPO further permits a momentum bias to be applied to the inner-loop initialization.

- We optimize a PPO baseline through extensive hyperparameter sweeps (total of 38,400 agent trained) on subsets of Brax (6 tasks) (Freeman et al., 2021), Jumanji (4 tasks) (Bonnet et al., 2024), and MinAtar (4 tasks) (Young & Tian, 2019). We open-source the sweep database files to facilitate future research against strongly tuned baselines.

- We perform three lightweight outer-PPO grid searches on non-unity outer learning rates, outer Nesterov momentum and biased-initialization, each addressing questions 1, 2 and 3 respectively.

- We evaluate the outer-PPO methods against the baseline, using 64 seeds per task over the 14 different tasks. We find non-unity outer learning rates to yield the greatest improvement (5-10%) on both Brax and Jumanji. Outer Nesterov also improves performance on Brax and Jumanji. Biased initialization achieves a moderate improvement on Jumanji alone. No method improves over the baseline on MinAtar.

- Given the stated empirical results we conclude the *negative* for questions 1, 2 and 3. Relaxing each of these PPO design choices can lead to consistent, statistically significant improvement of performance over at least one of the evaluated environment suites.

- We propose that practitioners able to experiment may explore non-unity outer learning rates given the simplicity (single hyperparameter) and consistent improvement achieved on Brax and Jumanjji.

## 2 BACKGROUND

### 2.1 REINFORCEMENT LEARNING

We consider the standard reinforcement learning formulation of a Markov decision process $\mathcal{M} = \langle \mathcal{S}, \mathcal{A}, \mathcal{T}, r, \gamma \rangle$, where $\mathcal{S}$ is the set of states, $\mathcal{A}$ is the set of actions, $\mathcal{T} : \mathcal{S} \times \mathcal{A} \to \Delta(\mathcal{S})$ is the state transition probability function, $r : \mathcal{S} \times \mathcal{A} \to \Delta(\mathbb{R})$ is the reward function, and $\gamma \in [0, 1]$ is the discount factor. We use the notation $\Delta(\boldsymbol{X})$ to denote the probability distribution over a set $\boldsymbol{X}$. The reinforcement learning objective is to maximize the expected return $\mathbb{E}_\pi[G_t] = \mathbb{E}_\pi[\sum_t \gamma^t r_t]$ given a *policy* $\pi : \mathcal{S} \to \Delta(\mathcal{A})$ defining the agent behavior. In actor-critic policy optimization the policy is explicitly represented as a parametric function $\pi : \mathcal{S} \times \boldsymbol{\theta}^\pi \to \Delta(\mathcal{A})$, and a *value function* $V : \mathcal{S} \times \boldsymbol{\theta}^V \to \mathbb{R}$ is employed to guide optimization. In deep RL (Mnih et al., 2015; Silver et al., 2017) neural networks are used for the policy and value functions, for ease of notation we consider $\boldsymbol{\theta} \in \mathbb{R}^{(d_\pi + d_V)}$ as the concatenation of the respective weight vectors.

### 2.2 PROXIMAL POLICY OPTIMIZATION

Proximal policy optimization was proposed by Schulman et al. (2017b), and has since become one of the most popular algorithms for on-policy reinforcement learning. At each iteration $k$ a dataset of transitions $\mathcal{D}_k$ is collected using policy $\pi(\boldsymbol{\theta}_k)$, and advantages $\hat{A}_k$ are estimated using generalized advantage estimation (GAE) (Schulman et al., 2018). The transition dataset and advantages are then used within an inner optimization loop, in which the policy parameters $\boldsymbol{\theta}^\pi$ are optimized with respect to a given surrogate objective along with the value parameters $\boldsymbol{\theta}^V$. Psuedocode for a single iteration of PPO is provided in algorithm 1, where INNEROPTIMIZATIONLOOP is defined in appendix A. The full algorithm updates parameters iteratively by $\boldsymbol{\theta}_{k+1} \leftarrow \text{PPOITERATION}(\boldsymbol{\theta}_k)$.

---

**Algorithm 1** Proximal policy optimization iteration

1: **function** PPOITERATION($\boldsymbol{\theta}$)
2:     Collect set of trajectories $\mathcal{D}$ by running policy $\pi(\boldsymbol{\theta})$
3:     Estimate advantages $\hat{A}$ with GAE.
4:     $\boldsymbol{\theta}^* \leftarrow \text{INNEROPTIMIZATIONLOOP}(\boldsymbol{\theta}, \mathcal{D}, \hat{A})$
5:     **return** $\boldsymbol{\theta}^*$
6: **end function**

---

PPO permits the use of any arbitrary surrogate objective, though it is most commonly associated with the *clipped objective* Schulman et al. (2017b) stated in equation 1.

$$L^\pi(\boldsymbol{\theta}^\pi) = \mathbb{E}_{s,a \sim \mathcal{D}_k}\left[\min\left(\rho(\boldsymbol{\theta}^\pi)\hat{A}, \ \text{clip}(\rho(\boldsymbol{\theta}^\pi), 1 - \epsilon, 1 + \epsilon)\hat{A}\right)\right] \tag{1}$$

Here $\rho(\boldsymbol{\theta}^\pi) = \frac{\pi(a|s)}{\pi_k(a|s)}$ is the ratio between our current policy $\pi$ and the behavior policy $\pi_k$, and $\epsilon$ is the clipping threshold. The value function is similarly optimized using either simple regression. $L^V(\boldsymbol{\theta}^V) = (V_{\boldsymbol{\theta}_k} - V_{\text{targ}})^2$ or the clipped objective defined in appendix A.

### 2.3 TRUST REGIONS

A *trust region* is a region surrounding an optimization iterate $\boldsymbol{\theta}_k$ within which we permit our algorithm to update the parameters to $\boldsymbol{\theta}_{k+1}$. In TRPO, a trust region surrounding the behavior parameters is explicitly defined as the region in parameter space $\boldsymbol{\theta} \in \Theta$ satisfying $\mathbb{E}_{s \sim \mathcal{D}_k}\left[D_{\text{KL}}\left(\pi(\boldsymbol{\theta}_k|s) \parallel \pi(\boldsymbol{\theta}|s)\right)\right] \leq \delta$. Optimizing subject to this constraint prevents large changes in the policy between successive iterations, and gives rise to a guarantee of *monotonic improvement*. Similarly, if the clipped surrogate objective of PPO is replaced with a KL penalty $L^\pi(\boldsymbol{\theta}) = \mathbb{E}_{s,a \sim \mathcal{D}_k}[\rho(\boldsymbol{\theta})\hat{A} - \beta D_{\text{KL}}\left(\pi(\boldsymbol{\theta}_k|s) \parallel \pi(\boldsymbol{\theta}|s)\right)]$, a trust-region is implicitly defined for some $\delta$. Both TRPO and PPO-KL approximate the natural policy gradient (Kakade, 2001), (Hsu et al., 2020); the steepest direction in the non-Euclidean geometry of policy space induced by the Fisher information metric.

Unlike the KL penalized surrogate, the clipped surrogate objective of equation 1 *does not* define a formal trust region. We can however define the region of non-zero gradients, with gradient defined as in equation 2.

$$\nabla_{\theta^{\pi}} L^{\pi}(\boldsymbol{\theta}^{\pi}) = \mathbb{E}_{s,a\sim\mathcal{D}_k} \left[ \hat{A} \nabla_{\boldsymbol{\theta}^{\pi}} \rho(\boldsymbol{\theta}^{\pi}) \cdot \mathbb{I} \left( |\rho(\boldsymbol{\theta}^{\pi}) - 1| \leq \epsilon \text{ or } (\rho(\boldsymbol{\theta}^{\pi}) - 1)\,\hat{A} \leq 0 \right) \right] \quad (2)$$

Here $\mathbb{I}(\cdot)$ is an indicator function that equals 1 if and only if the argument is true, and 0 otherwise. Whilst the subspace $\nabla_{\boldsymbol{\theta}} L^{\pi} \neq 0$ can be considered analogous to a trust region, it is possible to irreversibly step arbitrarily far beyond this region (Hsu et al., 2020). Nonetheless, where not ambiguous we shall abuse notation and refer to $\nabla_{\boldsymbol{\theta}} L^{\pi} \neq 0$ as the trust region of the clipped surrogate. Whilst not defining a formal trust region, the clipped objective enjoys theoretical motivation as a valid drift function in the mirror learning framework (Kuba et al., 2022), hence also benefits from monotonic improvement and convergence guarantees.

## 3 OUTER-PPO

In equation 3 we define the outer gradient of PPO.

$$\boldsymbol{g}^{O}(\boldsymbol{\theta}) = \text{PPOITERATION}(\boldsymbol{\theta}) - \boldsymbol{\theta} \quad (3)$$

The behavior parameter update of PPO $\boldsymbol{\theta}_{k+1} \leftarrow \text{PPOITERATION}(\boldsymbol{\theta}_k)$ can now be equivalently expressed as $\boldsymbol{\theta}_{k+1} \leftarrow \boldsymbol{\theta}_k + \boldsymbol{g}^{O}(\boldsymbol{\theta}_k)$. Evidently, PPO is exactly gradient ascent, with a constant learning rate $\sigma = 1$, on its outer gradients. With this simple result established, we propose a family of methods employing arbitrary optimizers on the PPO outer loop, denoted as *outer-PPO*. As an illustrating example, a comparison of standard PPO and outer-PPO with non-unity learning rates is provided in algorithms 2 and 3. We additionally propose a closely-related method for biasing the inner estimation loop using the prior (outer) trajectory, denoted as *biased initialization*.

### 3.1 OUTER LEARNING RATES

Varying the outer learning rate scales the update applied to the behavior parameters, as defined in algorithm 3 and illustrated in figure 1. The behavior of scaling the outer gradient can not be directly recovered by varying the PPO hyperparameters.

| **Algorithm 2** Standard PPO | **Algorithm 3** Outer-LR PPO |
|---|---|
| Input: $\boldsymbol{\theta}_0$ (parameters) | Input: $\boldsymbol{\theta}_0$ (parameters), $\sigma$ (outer learning rate) |
| 1: **for** $k = 0, 1, 2, \ldots$ **do** | 1: **for** $k = 0, 1, 2, \ldots$ **do** |
| 2: $\quad \boldsymbol{\theta}^* \leftarrow \text{PPOITERATION}(\boldsymbol{\theta}_k)$ | 2: $\quad \boldsymbol{g}_k^O \leftarrow \text{PPOITERATION}(\boldsymbol{\theta}_k) - \boldsymbol{\theta}_k$ |
| 3: $\quad \boldsymbol{\theta}_{k+1} \leftarrow \boldsymbol{\theta}^*$ | 3: $\quad \boldsymbol{\theta}_{k+1} \leftarrow \boldsymbol{\theta}_k + \sigma \boldsymbol{g}_k^O$ |
| 4: **end for** | 4: **end for** |

An outer learning rate $\sigma < 1$ interpolates between the behavior parameters $\boldsymbol{\theta}_k$ and inner-loop solution $\boldsymbol{\theta}_k^*$, encoding a lack of trust in the outer gradient estimation. Whilst the magnitude of the outer gradient can be reduced by varying hyperparameters, such as the clipping $\epsilon$ or number of inner loop iterations, the outer gradients are inherently noisy due to stochastic data collection and inner-loop optimization. PPO is additionally able to irreversibly escape its clipping boundary (Engstrom et al., 2020), and can drift far from the behavior policy given sub-optimal surrogate objective parameters. Finally, whilst by monotonic improvement guarantees we can assume $\boldsymbol{\theta}_k^*$ to define an equal or superior policy to $\boldsymbol{\theta}_k$, the non-linear map from parameters to policy and non-convex surrogate objective imply we cannot assume performance monotonically improves on the linear interpolation between these points. These effects motivate the exploration of methods that attenuate the outer updates, irrespective of the outer gradient magnitude. In contrast, a learning rate $\sigma > 1$ amplifies the update vector, encoding confidence in its direction. Whilst the outer gradient magnitude could be increased by varying the PPO hyperparameters, in particular $\epsilon$, increasing the size of the trust region may lead the policy to drift to beyond the region of policy space where the dataset $\mathcal{D}_k$ collected with policy $\pi_k$ can be considered representative of the environment dynamics, motivating the amplification of well-estimated outer gradients over increases to trust region size.

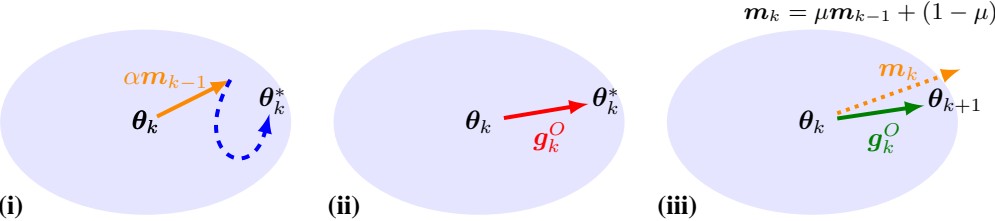

(a) **Outer-Nesterov PPO**. At each iteration Outer-Nesterov PPO estimates an outer gradient $\boldsymbol{g}_k^O$, updates the momentum $\boldsymbol{m}_k$, and steps the parameters using the Nesterov momentum update. The momentum step therefore *precedes* the construction of the following trust region, since it defines the following behavior policy $\pi(\boldsymbol{\theta}_{k+1})$.

(b) **Biased initialization**. Each iteration commences with a momentum step (solid orange); the inner optimization (blue dashed) is therefore initialized at $\boldsymbol{\theta}_k + \alpha \boldsymbol{m}_{k-1}$. The momentum step therefore *occurs within* the trust region as the dataset $\mathcal{D}_k$ was collected prior, and the surrogate objective remains defined relative to $\pi(\boldsymbol{\theta}_k)$.

Figure 2: **Comparison of Nesterov-PPO and biased initialization**.

## 3.2 MOMENTUM

Whilst permitting novel behavior, outer-LR PPO still only exploits information from a single PPO iteration when updating the parameters. Applying momentum breaks this design choice; instead of directly updating the parameters with the scaled outer gradient $\sigma \boldsymbol{g}_k^O$, we update using the Nesterov momentum rule as in algorithm 4 and illustrated in figure 2a.

---

**Algorithm 4** Outer-Nesterov PPO

1: Input: $\boldsymbol{\theta}_0$ (parameters), $\sigma$ (learning rate), $\mu$ (momentum factor)
2: $\boldsymbol{m}_0 \leftarrow \boldsymbol{0} \in \mathbb{R}^d$
3: **for** $k = 0, 1, 2, \ldots$ **do**
4: $\quad \boldsymbol{g}_k^O \leftarrow \text{PPOITERATION}(\boldsymbol{\theta}_k) - \boldsymbol{\theta}_k$
5: $\quad \boldsymbol{m}_k \leftarrow \mu \boldsymbol{m}_{k-1} + \boldsymbol{g}_k^O$
6: $\quad \boldsymbol{\theta}_{k+1} \leftarrow \boldsymbol{\theta}_k + \sigma(\boldsymbol{m}_k + \mu \boldsymbol{g}_k^O)$
7: **end for**

---

In supervised learning momentum is motivated using pathological curvature, and the ability to 'build up speed' (Sutskever et al., 2013). Given that the outer gradient is the solution to a surrogate objective, we do not anticipate pathological curvature presenting to the outer optimizer. However, similar to learning rates $\sigma > 1$ the increase in effective learning rate of momentum may assist in learning. Momentum can also be motivated here using resilience to noise; since any given collected dataset will be noisy, the outer gradient is also noisy. As using a learning rate $\sigma < 1$ corresponded to a lack of trust in any given outer gradient, using momentum corresponds to a smoothing process, where we at no point solely trust a single outer gradient to be accurate.

## 3.3 BIASED INITIALIZATION

Outer-PPO Nesterov applies a momentum-based update to the outer loop of PPO. This update occurs *before* the successive iteration's dataset $\mathcal{D}_{k+1}$ is collected, hence the momentum directly determines

the behavior parameters $\pi_{k+1}$ for the following surrogate objective. Beyond the effects of stateful inner-loop optimizers such as Adam (Kingma & Ba, 2014), each outer gradient estimation is independent of the prior trajectory. In contrast we propose biased initialization to apply an outer momentum-based update *after* data is collected, hence *inside* the following trust region problem as in algorithm 5, where $\boldsymbol{m}_k = \mu \boldsymbol{m}_{k-1} + (1 - \mu) \boldsymbol{g}_k^O$ is the momentum vector, and in figure 2b.

---

**Algorithm 5** PPO iteration with biased initialization

---

1: **function** BIASEDPPOITERATION($\boldsymbol{\theta}, \boldsymbol{m}, \alpha$)
2:      Collect set of trajectories $\mathcal{D}$ by running policy $\pi(\boldsymbol{\theta})$
3:      Compute advantages $\hat{A}$.
4:      $\boldsymbol{\theta} \leftarrow \boldsymbol{\theta} + \alpha \boldsymbol{m}$
5:      $\boldsymbol{\theta}^* \leftarrow$ INNEROPTIMIZATIONLOOP($\boldsymbol{\theta}, \mathcal{D}, \hat{A}$)
6:      **return** $\boldsymbol{\theta}^*$
7: **end function**

---

Biased initialization bears a strong similarity to the conjugate gradient initialization employed in Hessian-free optimization (Martens, 2010). The primary motivation for such techniques would be to better estimate the update vector in a given budget of inner-loop iterations.

## 4 EXPERIMENTS

### 4.1 EVALUATION PROCEDURE

We experiment on subsets of the Brax (Freeman et al., 2021), Jumanji (Bonnet et al., 2024), and MinAtar (Young & Tian, 2019) environment suites, selected as diverse examples of continuous and discrete control problems. We employ the absolute evaluation procedure recommended by Colas et al. (2018) and Gorsane et al. (2022). Absolute evaluation entails intermediate evaluations during training and a final, large-scale evaluation using the best policy identified to give the 'absolute' performance. We train with a budget of $1 \times 10^7$ transitions, perform 20 intermediate evaluations, and conduct final evaluation using 1280 episodes.

Recognizing the hyperparameter sensitivity of deep reinforcement learning (Hsu et al., 2020; Engstrom et al., 2020; Andrychowicz et al., 2020a), we commit significant resources to establishing a strong PPO baseline and fair evaluation. We sweep for a budget of 600 trials per task using the tree-structured Parzen estimator (Bergstra et al., 2011; Watanabe, 2023). Each trial is the mean of 4 agents, trained using seeds randomly sampled from $[0, 10000]$, for a total of 2400 agents trained per task during baseline tuning. A total of 11 hyperparameters are tuned, each with extensive ranges considered. Full descriptions of the hyperparameter sweep ranges, and the optimal values identified are provided in appendix C.

After hyperparameter tuning a final 64 agents are trained per environment task, where the set of evaluation seeds is non-overlapping with seeds used for hyperparameter tuning. To compare methods we aggregate performance over the tasks of an environment suite following the procedure recommended by Agarwal et al. (2021), normalizing with the min/max return found for each task across all trained agents (including sweep agents), a table of which is presented in appendix D.

### 4.2 DEFINED EXPERIMENTS

We consider the three outer-PPO methods defined in section 3; outer-LR, outer-Nesterov and biased initialization, addressing questions 1, 2, and 3 respectively. The outer-PPO methods are grid searched using increments of 0.1 for all hyperparameters. Outer-LR has a single hyperparameter; outer learning rate $\sigma$, which is swept over the range $[0.1, 4.0]$ (40 trials). Nesterov-PPO two hyperparameters; $\sigma$ $[0.1, 1.0]$ and momentum factor $\mu$ $[0.1, 0.9]$ (90 trials). Biased initialization also has two hyperparameters; bias learning rate $\alpha$ $[0.1, 1.0]$, bias momentum $\mu$ $[0.0, 0.9]$ (100 trials). The base PPO hyperparameters are frozen from the baseline sweep up to the 500th trial, such that no method is tuned using a budget greater than the 600 trials used by the baseline. The optimal hyperparameters identified for each sweep are provided in the figures of appendix E.

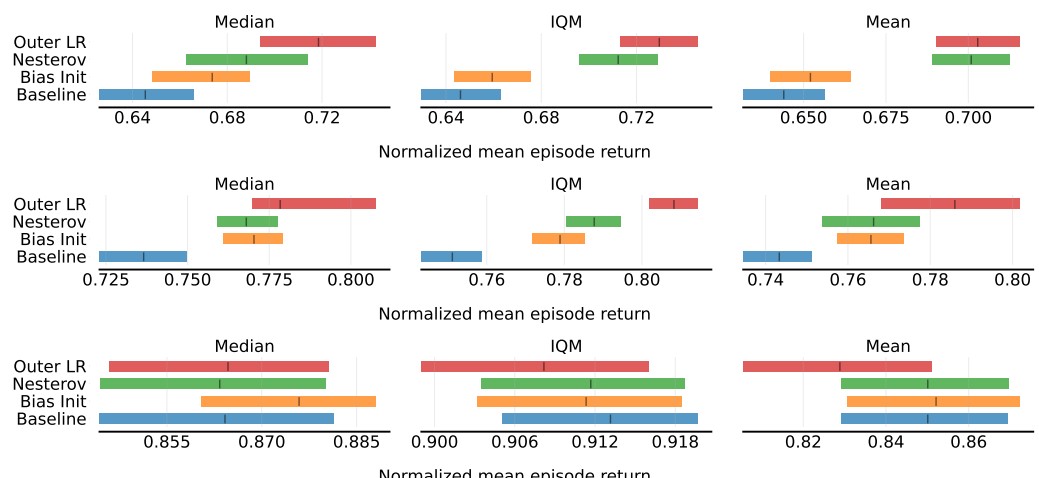

Figure 3: **Aggregate point estimates** for Brax (upper), Jumanji (center), and MinAtar (lower). Optimal hyperparameters *per-environment* are used. Normalized to task min/max across all experiments.

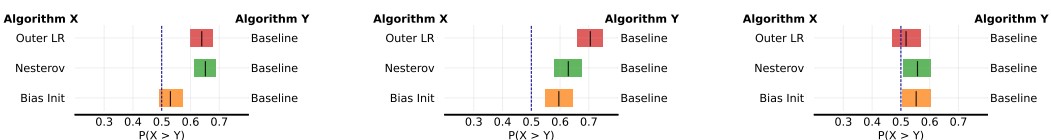

Figure 4: **Probability of improvement** for Brax (left), Jumanji (center), and MinAtar (right). Optimal hyperparameters *per-environment* are used. Normalized to task min/max across all experiments.

## 5 RESULTS

### 5.1 EMPIRICAL PERFORMANCE

We first consider the performance of the three outer-PPO methods, where the optimal hyperparameters identified from the grid sweeps *per-environment* are employed. In figures 3 and 4 we present the aggregate point estimates and probability of improvement. Further results including sample efficiency curves are provided in appendix D.

**Aggregate point estimates**. Outer-LR demonstrates a statistically significant improvement over the PPO baseline on Brax and Jumanji for all point estimates considered (median, IQM, mean, optimality gap). Outer-Nesterov also demonstrates enhanced performance on Brax and Jumanji; this improvement is less substantial than that of outer-LR but remains statistically significant on all point estimates aside from the Brax median. Biased initialization is the weakest of the outer-PPO instantiations, with minor improvements lacking statistical significance on Brax and moderate but significant improvements on Jumanji. No method improves over baseline on MinAtar.

**Probability of improvement**. All methods have a probability of improvement (over baseline) greater than 0.5. In most cases this improvement is statistically significant, aside from biased initialization on Brax and outer-LR on MinAtar. Notably, outer-LR has a probability of improvement greater than 0.6 on Brax and greater than 0.7 on Jumanji.

### 5.2 HYPERPARAMETER SENSITIVITY

In the results of figures 3 and 4, the optimal hyperparameters from each *per-environment* outer-PPO grid search are used. We now consider the sensitivity of outer-PPO to these hyperparameters. In figures 5, 6, and 7 we present the return, *normalized across each environment suite*, as a function of the sweep hyperparameters for outer-LR, outer-Nesterov and biased initialization. Normaliza-

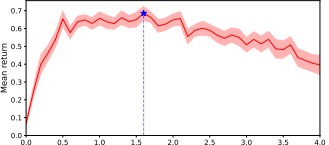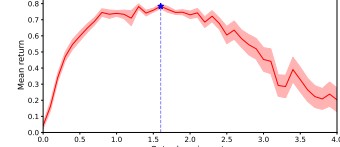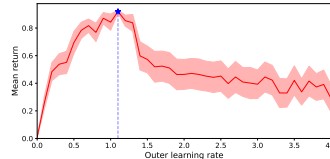

Figure 5: **Outer-LR hyperparameter sensivity.** Mean normalized return across the Brax (left), Jumanji (center), MinAtar (right) tasks as a function of outer learning rate $\sigma$. Mean of 4 seeds plotted with standard error shaded. Normalized to task min/max across all experiments. Common outer learning rate used to define the $x$-axis, with task-specific base PPO hyperparameters.

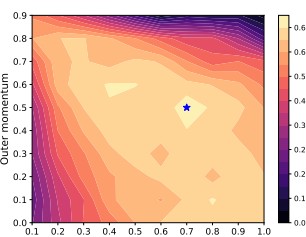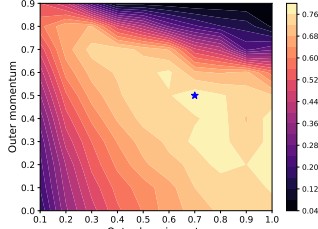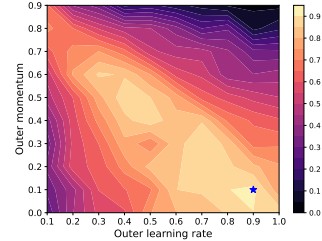

Figure 6: **Outer-Nesterov hyperparameter sensitivity.** Mean normalized return across the Brax (left), Jumanji (center), MinAtar (right) tasks as a function of outer learning rate $\sigma$ and outer momentum $\mu$. Mean of 4 seeds plotted. Normalized to task min/max across all experiments. Common outer hyperparameters used to define the grid, with task-specific base PPO hyperparameters.

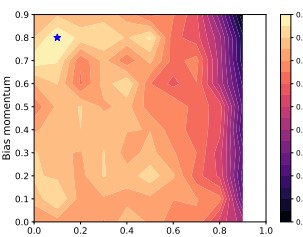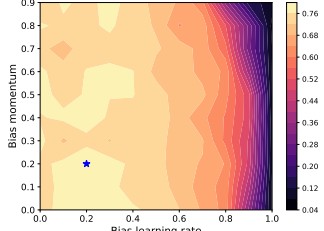

Figure 7: **Biased initialization hyperparameter sensitivity.** Mean normalized return across the Brax (left), Jumanji (center), MinAtar (right) tasks as a function of bias init learning rate $\alpha$ and bias momentum $\mu$. Mean of 4 seeds plotted. Normalized to task min/max across all experiments. Common outer hyperparameters used to define the grid, with task-specific base PPO hyperparameters.

tion is again performed using the extreme values presented in appendix D. Analogous plots for the individual tasks are provided in appendix E.

**Outer learning rate.** When normalized across all tasks, Brax has low sensitivity to outer learning rate. The range of values $\sigma \in [0.8, 2.0]$ has comparable performance to the peak located at $\sigma = 1.6$. Notably, performance is not greatly reduced when using values up to $\sigma = 3.0$. Jumanji again exhibits near optimal-performance over a broad range of values $\sigma \in [0.5, 2.2]$, with the peak again located at $\sigma = 1.6$. Unlike in Brax, performance on Jumanji is greatly diminished for values $\sigma > 2.5$. MinAtar has a sharp peak in performance around standard PPO ($\sigma = 1.0$), with a rapid decrease in performance for values greater than this.

**Nesterov.** All three suites have a ridge-like trend in normalized performance, with poor performance where $\sigma$ and $\mu$ are both small or both large. Both Brax and Jumanji have their peak at $(\sigma, \mu) = (0.7, 0.5)$, with a relatively broad plateau of near-optimal performance. The peak of MinAtar is at $(\sigma, \mu) = (0.9, 0.1)$, with a narrow ridge of near-optimal performance.

**Biased initialization.** The dominant trend on all three suites is decreasing normalized performance for large bias learning rate $\alpha$. The optima for all suites at either $\alpha = 0.1$ (Brax, MinAtar) or $\alpha = 0.2$ (Jumajji). There is comparably little variation with respect to bias momentum $\mu$, with the suite optima dispersed through the available range. Jumanji has a broader region of near optimal performance than Brax or MinAtar, covering $\alpha < 0.4$.

## 6 DISCUSSION

We now reflect on the questions posed in section 1. The PPO baselines in this work were tuned aggressively for each task, greatly increasing the confidence in the experimental findings. Given the baseline strength, and performance demonstrated in figures 3 and 4, we conclude in the *negative* for all three questions as evidenced by:

**Q1.** Varying the outer learning rate leads to an statistically significant increase on all point estimates on Brax and Jumanji, with corresponding increases to probability of improvement.

**Q2.** Employing Nesterov momentum on the outer loop, with outer learning rate attenuation, achieves statistically significant increases to all point estimates on Brax and Jumanji. We also observe a statistically significant probability of improvement on all three suites.

**Q3.** Momentum-biased initialization achieves statistically significant increase on all point estimates on Jumanji, with a probability of improvement of 0.6 on this suite.

**Common hyperparameters.** The sensitivity plots in figures 5, 6 demonstrate robust normalized performance across the Brax and Jumanji suites for outer-LR and outer-Nesterov. However, they do not indicate any significant increase in normalized return could be achieved over standard PPO for a set of common hyperparameters shared across a suite. To achieve the improved aggregate metrics in figure 3 it was necessary to use task-specific hyperparameters. We do however emphasize the aggressive, task-specific, tuning of the baseline, and view the robustness of normalized return across a range of hyperparameters as a strength of the methods.

**Task-specific hyperparameters.** Task-specific hyperparameter sensitivity plots are provided in appendix E. For outer-LR the optimal per-task values for $\alpha$ range between 0.5 (corresponding to cautious updates) and 2.3 (corresponding to confident updates). That values of $\alpha$ up to 2.3 can be optimal is surprising, as an $\alpha$ greater than unity directly violates the trust region established by our previous behavior policy. This precludes the provable monotonic improvement of PPO (Kuba et al., 2022); by stepping beyond the trust region we may in principle select a policy that is worse than the previous. For outer-Nesterov co-varying $\sigma$ with $\mu$ can be understood through the effective learning rate $\sigma/(1 - \mu)$. The task-specific effective learning rate varies from 0.7 to 2.3. Lastly, for biased initialization the sharp peaks in performance on Brax tasks suggest the method suffers from high variance on this suite, hence the hyperparameters selected may not be optimal in expectation. On Jumanji the method is significantly less hyperparameter sensitive as evidenced by the smooth contours, providing an explanation for the performance gap observed between these suites.

**MinAtar results.** No outer-PPO method improved over baseline on MinAtar. We comment that our baseline results are much stronger than other works (Lu et al., 2022), and are approaching the mathematical maxima of these tasks as defined by the gymnax library (Lange, 2022). We further add that other works committing substantial resources to baseline tuning on MinAtar have struggled to achieve improvements on the suite Jesson et al. (2023). Furthermore, the hyperparameter sensitivity plots in figures 5, 6 and 7 demonstrate all methods achieve peak normalized return greater than 0.9 on MinAtar. Since here we are normalizing to the maximum performing agents across all sweeps, this indicates there is less variance in the optimal performance of MinAtar compared to Brax and Jumanji with peak normalized returns around 0.7 and 0.8 respectively. A final explanation for the failure to surpass baseline on MinAtar could be 'brittle' base hyperparameters, not suited to the modified dynamics introduced by outer-PPO, supported by the sharp peak observed in outer-LR and concentration of performance in outer-Nesterov about standard PPO in figures 5 and 6.

**Limitations.** We identify two core limitations to this work; the *fixed transition budget* and the *lack of co-optimization* of base and outer-PPO hyperparameters. We only consider a timestep budget of

$1 \times 10^7$ transitions. Whilst sample efficiency plots are provided in appendix D the hyperparameters have not been tuned to maximize performance in the data-limited regime. Furthermore, we do not consider the asymptotic performance for larger transition budgets, where it is possible the improvement achieved by outer-PPO methods may be diminished. With respect to co-optimization, given the dependence of the outer gradients on the base hyperparameters there is undoubtedly significant interaction between these and the outer-PPO hyperparameters. Exploring these interactions would yield better understanding and potentially improved performance. We additionally highlight the presence of learning rate annealing on the inner Adam instances in all experiments. This implies the outer gradients tend to zero, the implications of which we do not explore in this work.

## 7 RELATED WORK

The usage of the difference between initial parameters and those after gradient-based optimization as a 'gradient' has been explored for meta-learning in the Reptile algorithm (Nichol et al., 2018). Reptile aims to find an initialization that can be quickly fine-tuned across a distribution of tasks. Unlike outer-PPO, which applies this idea within a single RL task, Reptile performs gradient steps on different supervised learning tasks to determine the 'Reptile gradient'. One could interpret outer-PPO as performing serial Reptile whereby each sampled task is the next PPO iteration alongside the collected dataset.

Whilst to the best of our knowledge we are the first to apply momentum to the outer loop of PPO, momentum-based optimizers such as RMSProp Tieleman & Hinton (2012) and Adam Kingma & Ba (2014) are commonly applied in other areas of RL. Recent work has examined the interaction of momentum based optimizers and RL objectives. Bengio et al. (2021) identify that a change in objective (such as by updating a target network or dataset), may lead to momentum estimates anti-parallel to the current gradient thereby hindering progress, and propose a correction term to mitigate this effect. Asadi et al. (2023) propose to reset the momentum estimates periodically throughout training and demonstrate improved performance on the Atari Learning Environment Bellemare et al. (2012) with Rainbow Hessel et al. (2017) doing so. However, none of these approaches focuses on PPO specifically, and instead address temporal difference learning or value based-methods.

Lastly, the biased initialization explored in this work is similar to the conjugate gradient initialization technique employed in hessian-free optimization Martens (2010), although this used only the prior iterate and not a momentum vector. Hessian-free optimization can be considered a supervised learning version of TRPO (Schulman et al., 2017a).

## 8 CONCLUSION

In this work, we introduced outer-PPO, a novel perspective of proximal policy optimization that applies arbitrary gradient-based optimizers to the outer loop of PPO. We posed three key research questions regarding the optimization process in PPO and conducted an empirical investigation across 14 tasks from three environments suites. Our experiments revealed that non-unity learning rates and momentum in the outer loop both yielded statistically significant performance improvements across a variety of evaluation metrics in the Brax and Jumanji environments, with gains ranging from 5-10% over a heavily tuned PPO baseline. Biased initialization provided improvements upon the baseline on Jumanji tasks but not Brax.

The most immediate direction for future research would be the exploration of interactions between base hyperparameters and outer-PPO hyperparameters. Since the optimal base hyperparameters may be unsuited to the modified dynamics of outer-PPO, the co-optimization of hyperparameters may yield performance improvements and deeper understanding of the method. Other possible future directions include the use of outer-PPO with alternatives to the clipped surrogate loss function, such as KL-penalized PPO Hsu et al. (2020) or discovered policy optimization Lu et al. (2022), and the use of adaptive optimizers on the outer loop such as RMSProp or Adam. Indeed, an 'outer' variant of many dual-loop RL algorithms can be defined, and we hope that this work will stimulate further research into optimizing RL algorithms through more sophisticated outer-loop strategies.

## ACKNOWLEDGMENTS

This work was supported by a Turing AI World-Leading Researcher Fellowship G111021. Research supported with Cloud TPUs from Google's TPU Research Cloud (TRC)

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

# A   FURTHER DETAILS ON PPO

## A.1   INNER OPTIMIZATION LOOP

---

**Algorithm 6** PPO Inner Optimization Loop

---

1: **Input:** $\boldsymbol{\theta}$ (initial parameters), $\mathcal{D}$ (collected trajectories), $\hat{A}$ (estimated advantages)
2: $\boldsymbol{\theta}^\pi, \boldsymbol{\theta}^V \leftarrow \boldsymbol{\theta}$
3: **for** epoch $i = 1, 2, \ldots, N$ **do**
4:     Shuffle $(\mathcal{D}, \hat{A})$ and create $M$ minibatches $\{(\mathcal{D}_1, \hat{A}_1), (\mathcal{D}_2, \hat{A}_2), \ldots, (\mathcal{D}_M, \hat{A}_M)\}$
5:     **for** $j = 1, 2, \ldots, M$ **do**
6:         $\boldsymbol{\theta}^\pi \leftarrow \boldsymbol{\theta}^\pi + \eta \nabla_{\boldsymbol{\theta}^\pi} L^\pi(\boldsymbol{\theta}^\pi, \mathcal{D}_j, \hat{A}_j)$
7:         $\boldsymbol{\theta}^V \leftarrow \boldsymbol{\theta}^V + \eta \nabla_{\boldsymbol{\theta}^V} L^V(\boldsymbol{\theta}^V, \mathcal{D}_j, \hat{A}_j)$
8:     **end for**
9: **end for**
10: $\boldsymbol{\theta} \leftarrow \boldsymbol{\theta}^\pi, \boldsymbol{\theta}^V$
11: **Return:** $\boldsymbol{\theta}^* \leftarrow \boldsymbol{\theta}$

---

Algorithm 6 describes the inner optimization loop of proximal policy optimization, where $L^\pi$ and $L^V$ are defined in equations 1 and 4 respectively. For notational ease this presentation is slightly simplified. Typically, instead of the gradient ascent steps taken in lines 5 and 6 typically each of $\boldsymbol{\theta}^\pi$ and $\boldsymbol{\theta}^V$ are optimized using independent instances of Adam (Kingma & Ba, 2014), with potentially distinct learning rates $\eta^\pi \neq \eta^V$. In this work we use Adam for the inner-loop optimization, following standard best-practice (Schulman et al., 2017b; Engstrom et al., 2020).

## A.2   CLIPPED VALUE OBJECTIVE

$$L^V(\boldsymbol{\theta}^V) = \max\left[ \left(V_{\boldsymbol{\theta}_k} - V_{\text{targ}}\right)^2, \left(\text{clip}\left(V_{\boldsymbol{\theta}_k}, V_{\boldsymbol{\theta}_{k-1}} - \varepsilon, V_{\boldsymbol{\theta}_{k-1}} + \varepsilon\right) - V_{\text{targ}}\right)^2 \right] \tag{4}$$

## B   IMPLEMENTATION DETAILS

We implement our experiments using the JAX-based Stoix library (Toledo, 2024). Our implementation is such that several seeds can be trialed / evaluated simultaneously for the same hyperparameters using a single device. We used Google Cloud TPU (v4-8) for these experiments. We used the gymnax Lange (2022) library implementation of MinAtar.

Table 1: PPO implementation details employed in this work as identified by Huang et al. (2022).

| Implementation Detail | Applied |
|---|---|
| Orthogonal Initialization | Yes |
| Adam Optimizer's Epsilon | Yes |
| Learning Rate Annealing | Yes |
| Generalized Advantage Estimation (GAE) | Yes |
| Mini-batch Updates | Yes |
| Normalization of Advantages | Yes |
| Clipped Surrogate Objective | Yes |
| Value Function Loss Clipping | Yes |
| Entropy Bonus | No |
| Global Gradient Clipping | Yes |
| Separate Networks | Yes |
| Observation Normalization | Yes |
| Reward Scaling | Yes |
| Reward Clipping | No |
| Normal Distribution for Actions | Yes |
| State-independent Log Std | No |
| Independent Action Components | Yes |
| Action Clipping | No |
| Action TanH Transform | Yes |
| Observation Clipping | No |

## C   HYPERPARAMETERS

### C.1   SWEEP RANGES

The sweep ranges for baseline hyperparameter sweeps are presented in table 2.

Table 2: Sweep ranges for baseline hyperparameters.

| Parameter | Sweep Range |
|---|---|
| Parallel environments | $2^6$ to $2^{10}$ |
| Rollout | $2^2$ to $2^8$ |
| Num. epoch | 1 to 16 |
| Num. minibatch | $2^0$ to $2^6$ |
| Actor learning rate | $1 \times 10^{-5}$ to $1 \times 10^{-3}$ (log scale) |
| Critic learning rate | $1 \times 10^{-5}$ to $1 \times 10^{-3}$ (log scale) |
| Discount factor ($\gamma$) | 0.9 to 1.0 |
| GAE $\lambda$ | 0.0 to 1.0 |
| Clip $\epsilon$ | 0.1 to 0.5 |
| Max gradient norm | 0.1 to 5.0 |
| Reward scaling | 0.1 to 100 (log scale) |

## C.2 OPTIMAL VALUES

The optimal values identified by the baseline sweep, up to trial 500, are included in table 3. These values are the 'base' hyperparameters used for outer-PPO methods.

Table 3: Optimal values from baseline sweep up to trial 500

| Task | Parallel env. | Rollout | Num. epoch | Num. m-batch | Actor lr | Critic lr | Discount $\gamma$ | GAE $\lambda$ | Clip $\epsilon$ | Max g. norm | Reward scale |
|---|---|---|---|---|---|---|---|---|---|---|---|
| ant | 128 | 8 | 2 | 32 | 3.0e-04 | 1.4e-04 | 0.98 | 0.70 | 0.21 | 4.85 | 0.14 |
| halfcheetah | 64 | 64 | 3 | 16 | 3.9e-04 | 4.4e-04 | 0.99 | 0.94 | 0.13 | 2.40 | 0.46 |
| hopper | 64 | 64 | 2 | 64 | 6.3e-04 | 3.6e-04 | 1.00 | 0.96 | 0.17 | 3.54 | 3.95 |
| humanoid | 256 | 64 | 4 | 64 | 1.0e-04 | 1.0e-04 | 0.98 | 0.89 | 0.34 | 3.30 | 0.14 |
| humanoidstandup | 64 | 64 | 3 | 32 | 3.0e-04 | 8.2e-04 | 0.99 | 0.98 | 0.10 | 4.65 | 0.35 |
| walker2d | 256 | 32 | 4 | 64 | 5.4e-04 | 8.2e-04 | 1.00 | 0.92 | 0.12 | 3.74 | 22.54 |
| asterix | 128 | 128 | 3 | 64 | 8.3e-04 | 2.1e-05 | 1.00 | 0.20 | 0.30 | 2.28 | 6.62 |
| breakout | 64 | 16 | 14 | 16 | 1.8e-04 | 1.2e-04 | 0.90 | 0.53 | 0.16 | 0.25 | 5.19 |
| freeway | 64 | 128 | 10 | 2 | 6.9e-04 | 1.3e-04 | 0.98 | 0.70 | 0.15 | 4.71 | 6.64 |
| space_invaders | 128 | 32 | 16 | 2 | 3.0e-05 | 1.1e-04 | 0.98 | 1.00 | 0.25 | 0.35 | 0.61 |
| game_2048 | 1024 | 8 | 9 | 32 | 4.9e-04 | 3.8e-04 | 0.99 | 0.04 | 0.28 | 2.56 | 0.13 |
| maze | 256 | 32 | 7 | 64 | 6.5e-04 | 4.3e-04 | 0.98 | 0.66 | 0.14 | 2.46 | 1.97 |
| rubiks_cube | 64 | 256 | 13 | 4 | 9.0e-04 | 2.2e-04 | 0.99 | 0.55 | 0.14 | 3.45 | 11.03 |
| snake | 1024 | 8 | 11 | 4 | 6.0e-04 | 6.0e-04 | 1.00 | 0.46 | 0.12 | 2.52 | 20.48 |

Table 4: Optimal hyperparameters per task for each outer-PPO method

| Task | Outer-LR | Outer-Nesterov | | Biased Initialization | |
|---|---|---|---|---|---|
| | $\sigma$ | $\sigma$ | $\mu$ | $\alpha$ | $\mu$ |
| Ant | 0.5 | 0.7 | 0.2 | 0.1 | 0.8 |
| HalfCheetah | 0.5 | 0.4 | 0.5 | 0.2 | 0.8 |
| Hopper | 1.5 | 0.9 | 0.4 | 0.5 | 0.8 |
| Humanoid | 1.9 | 0.5 | 0.7 | 0.1 | 0.4 |
| HumanoidStandup | 2.1 | 0.5 | 0.3 | 0.5 | 0.8 |
| Walker2d | 2.0 | 0.9 | 0.6 | 0.4 | 0.0 |
| 2048 | 1.3 | 0.8 | 0.4 | 0.3 | 0.9 |
| Snake | 2.3 | 1.0 | 0.4 | 0.7 | 0.5 |
| Rubik's Cube | 1.7 | 0.5 | 0.7 | 0.4 | 0.3 |
| Maze | 0.9 | 0.9 | 0.0 | 0.1 | 0.5 |
| Asterix | 1.1 | 0.6 | 0.5 | 0.1 | 0.4 |
| Breakout | 1.1 | 0.9 | 0.1 | 0.0 | 0.5 |
| Freeway | 1.6 | 0.9 | 0.3 | 0.2 | 0.5 |
| Space Invaders | 1.3 | 0.8 | 0.2 | 0.1 | 0.9 |

# D ADDITIONAL RESULTS

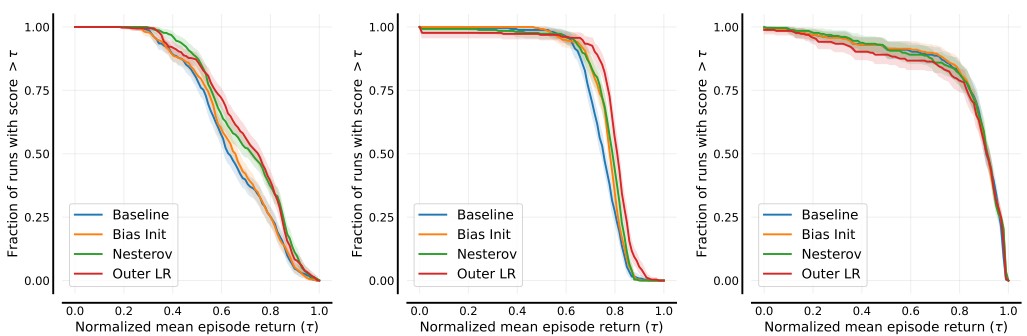

Figure 8: **Performance profiles for Brax (left), Jumanji (center), and MinAtar (right).** 6 / 4 / 4 tasks used from Brax / Jumanji / MinAtar respectively. For each task, agents are trained and evaluated using 64 seeds.

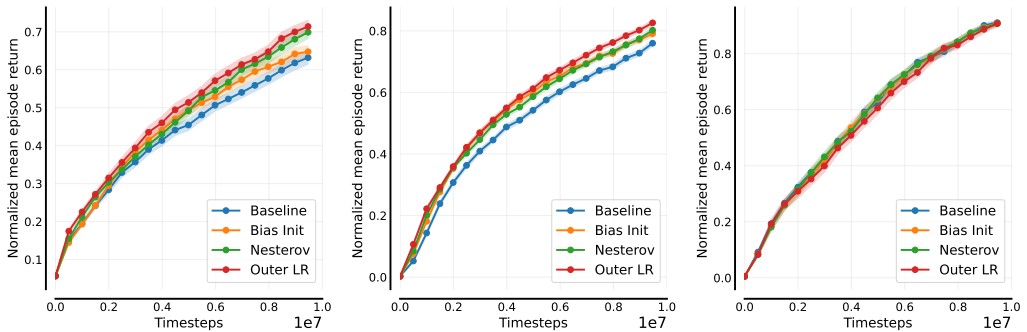

Figure 9: Sample efficiency curves for Brax (left), Jumanji (center), and MinAtar (right).

Table 5: Minimum and maximum returns used for normalization.

| Task | Min | Max |
|---|---|---|
| Ant | -2958.14 | 13466.48 |
| Halfcheetah | -587.37 | 7859.28 |
| Hopper | 21.03 | 3697.39 |
| Humanoid | 207.63 | 11851.71 |
| Humanoidstandup | 6686.00 | 71897.67 |
| Walker2d | -32.44 | 2558.61 |
| 2048 | 989.50 | 29084.63 |
| Snake | 0.00 | 92.55 |
| Rubiks Cube | 0.00 | 0.66 |
| Maze | 0.03 | 0.84 |
| Asterix | 0.30 | 64.46 |
| Breakout | 0.00 | 92.86 |
| Freeway | 0.00 | 66.13 |
| Space Invaders | 0.00 | 191.80 |

# E  SWEEP PERFORMANCES

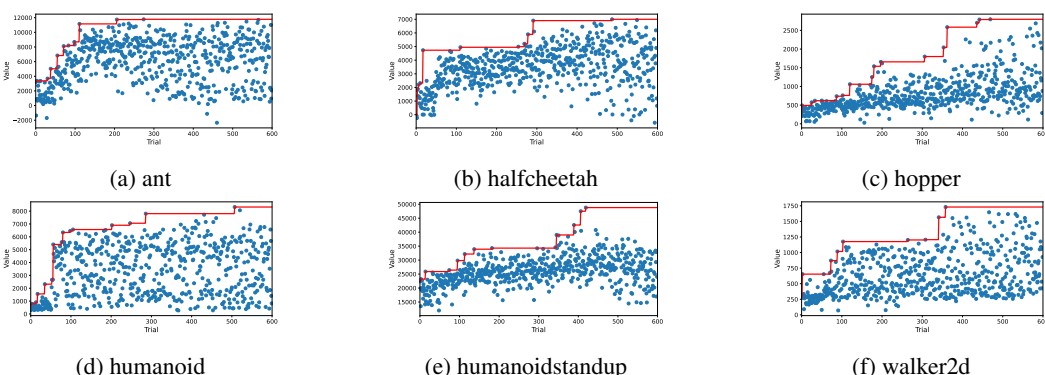

(a) ant  (b) halfcheetah  (c) hopper

(d) humanoid  (e) humanoidstandup  (f) walker2d

Figure 10: **Baseline sweep performance for Brax tasks.** $x$-axis is trial number, each trial represents a selection of hyperparameters selected by the Tree Parzen estimator. The $y$-axis is mean return achieved by the 4-seed trial. Red line represents cumulative maximum.

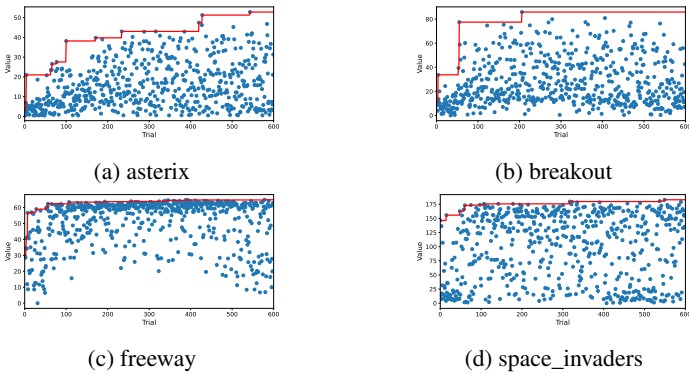

(a) asterix  (b) breakout

(c) freeway  (d) space_invaders

Figure 11: **Baseline sweep performance for MinAtar tasks.** $x$-axis is trial number, each trial represents a selection of hyperparameters selected by the Tree Parzen estimator. The $y$-axis is mean return achieved by the 4-seed trial. Red line represents cumulative maximum.

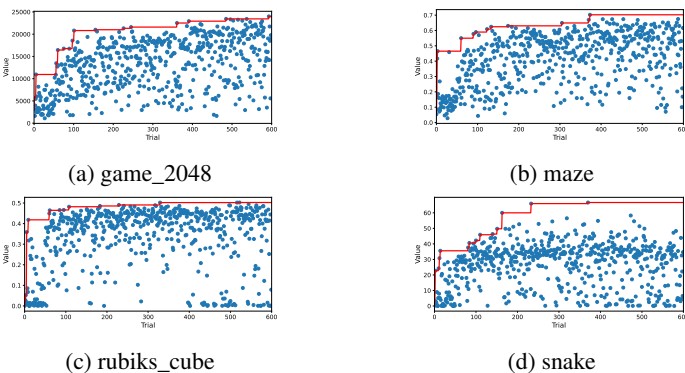

(a) game_2048  (b) maze

(c) rubiks_cube  (d) snake

Figure 12: **Baseline sweep performance for Jumajji tasks.** $x$-axis is trial number, each trial represents a selection of hyperparameters selected by the Tree Parzen estimator. The $y$-axis is mean return achieved by the 4-seed trial. Red line represents cumulative maximum.

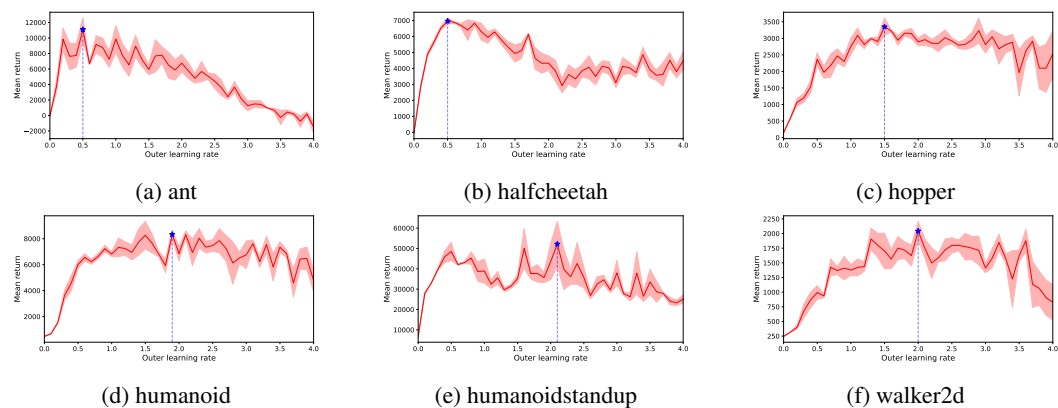

Figure 13: **Outer learning rate sweep performance for Brax tasks.** Mean of 4 seeds shown with standard error shaded. Optimal point marked with blue star.

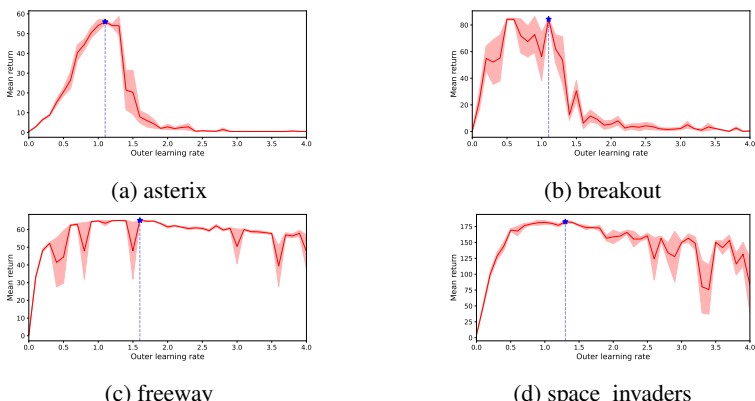

Figure 14: **Baseline sweep performance for MinAtar tasks.** Mean of 4 seeds shown with standard error shaded. Optimal point marked with blue star.

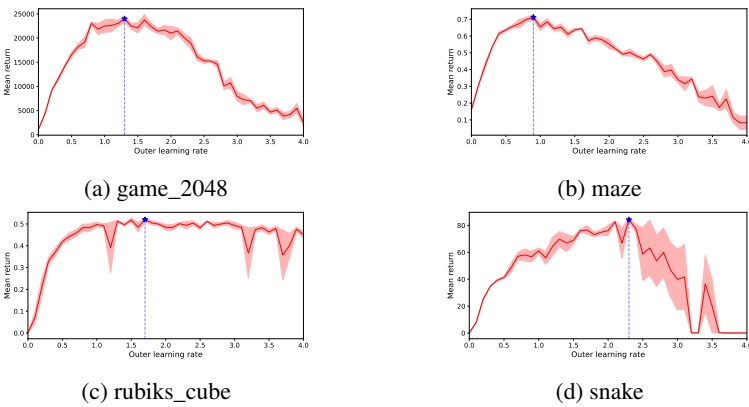

Figure 15: **Baseline sweep performance for Jumajji tasks.** Mean of 4 seeds shown with standard error shaded. Optimal point marked with blue star.

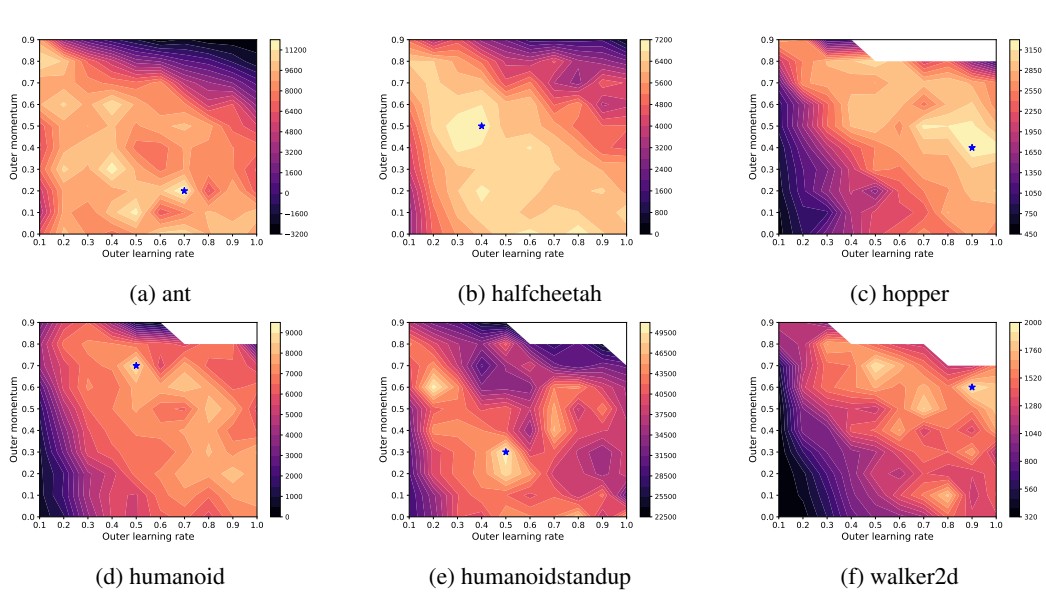

Figure 16: **Nesterov sweep performance for Brax tasks.** Contour plot of mean of 4 seeds. White regions resulted in numerical errors (NaN). Optimal point marked with blue star.

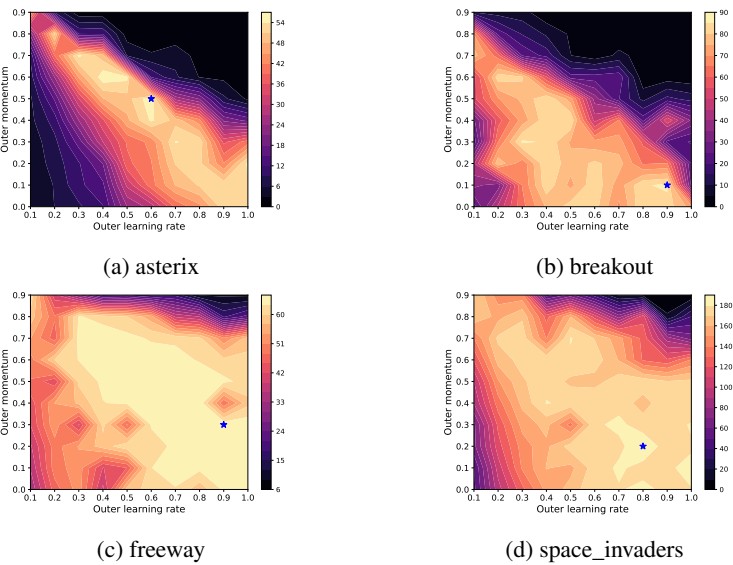

Figure 17: **Nesterov sweep performance for MinAtar tasks.** Contour plot of mean of 4 seeds. White regions resulted in numerical errors (NaN). Optimal point marked with blue star.

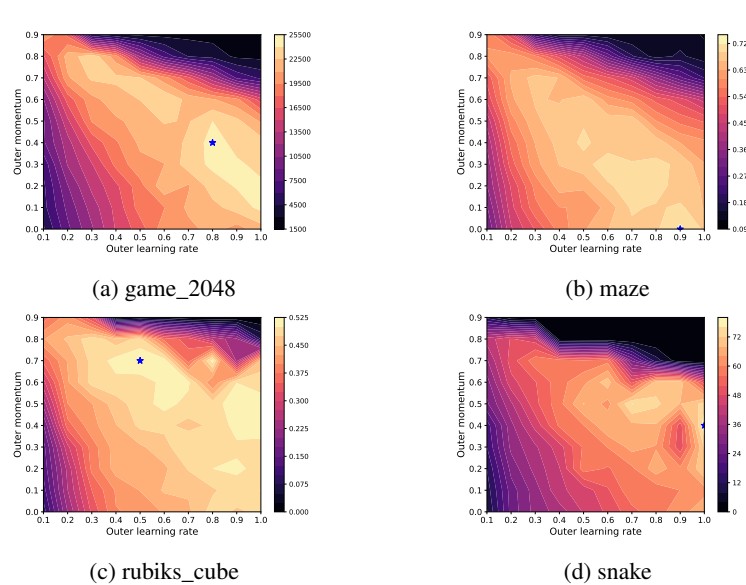

Figure 18: **Nesterov sweep performance for Jumanji tasks.** Contour plot of mean of 4 seeds. White regions resulted in numerical errors (NaN). Optimal point marked with blue star.

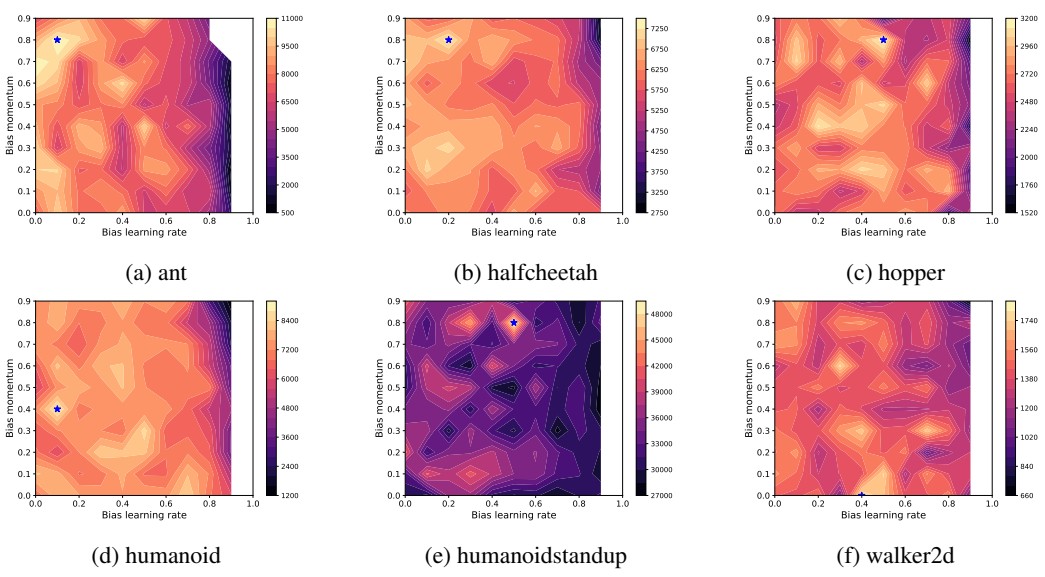

Figure 19: **Biased initialization sweep performance for Brax tasks.** Contour plot of mean of 4 seeds. White regions resulted in numerical errors (NaN). Optimal point marked with blue star.

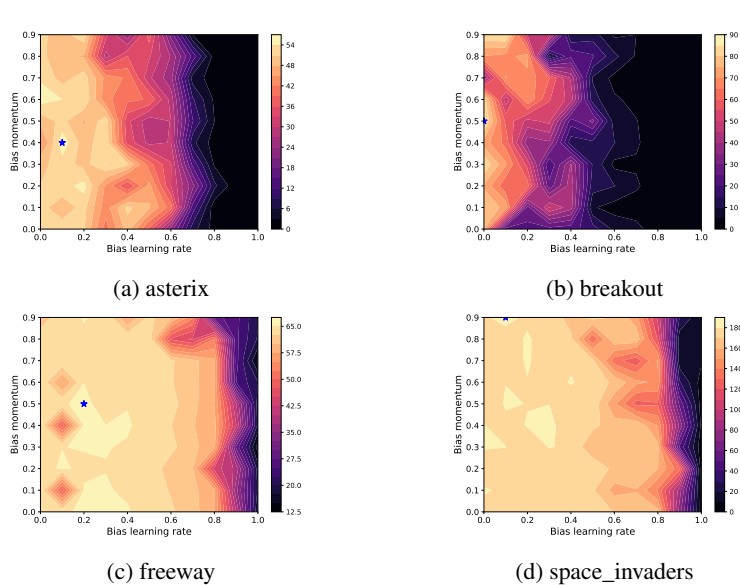

Figure 20: **Biased initialization sweep performance for MinAtar tasks.** Contour plot of mean of 4 seeds. White regions resulted in numerical errors (NaN). Optimal point marked with blue star.

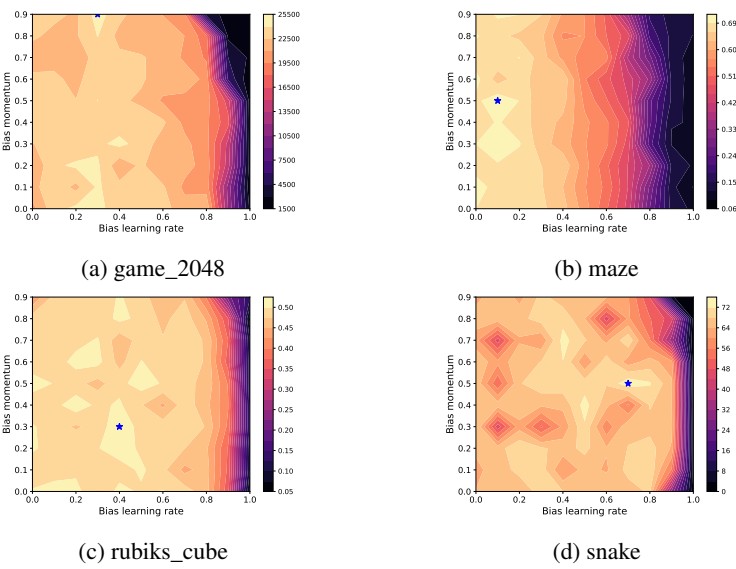

Figure 21: **Biased initialization sweep performance for Jumanji tasks.** Contour plot of mean of 4 seeds. White regions resulted in numerical errors (NaN). Optimal point marked with blue star.

# F INDIVIDUAL TASK PERFORMANCES

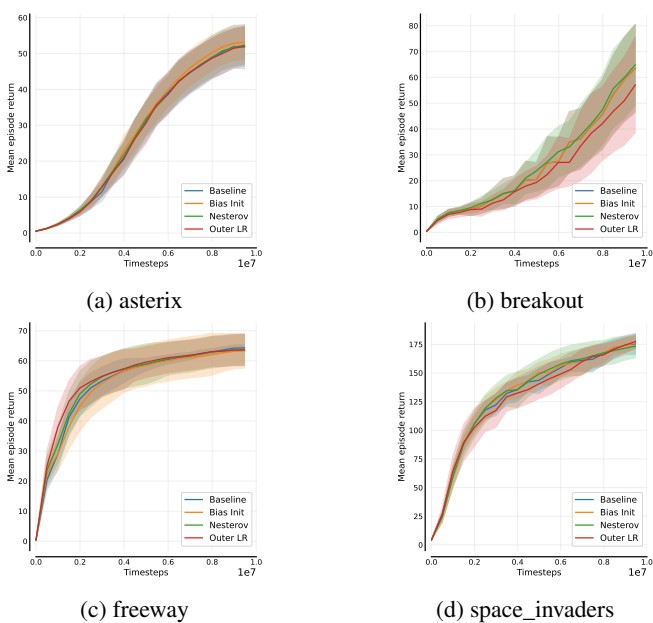

Figure 22: **Individual task performance for Brax.** For each task mean of 64 seeds is presented with standard deviation shaded.

Figure 23: **Individual task performance for MinAtar.** For each task mean of 64 seeds is presented with standard deviation shaded.

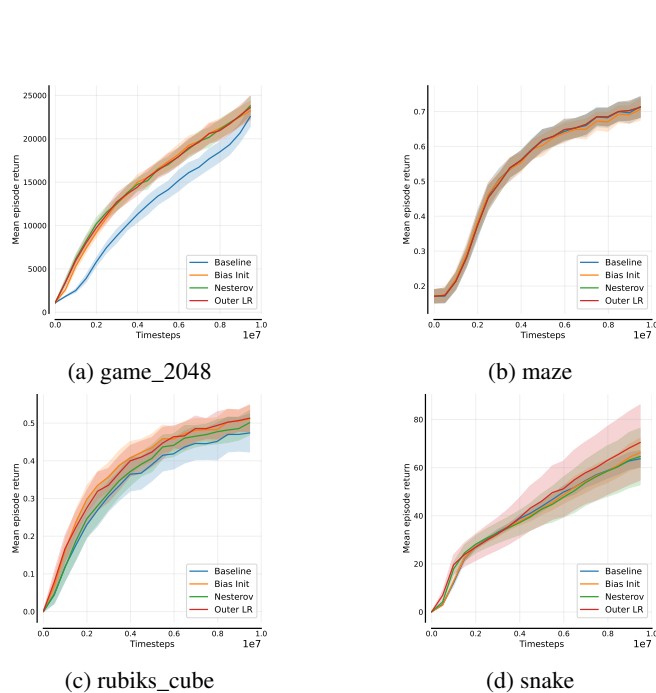

(a) game_2048

(b) maze

(c) rubiks_cube

(d) snake

Figure 24: **Individual task performance for Jumanji.** For each task mean of 64 seeds is presented with standard deviation shaded.

## G    CAN STANDARD PPO RECOVER OUTER-PPO?

Here we discuss that outer-PPO introduces novel behavior that cannot be recovered through variation of standard PPO hyperparameters. We focus our discussion on clipping $\epsilon$ and inner learning rates $\eta$ as highly influential hyperparameters, but similar arguments can be made for other hyperparameters.

### G.1    CLIPPING $\epsilon$

The clipping $\epsilon$ defines the policy ratio $\rho(\boldsymbol{\theta}^\pi)$ beyond which the loss is clipped, hence the gradients are zero. Intuitively, increasing $\epsilon$ increases the size of the trust region within which we seek to restrict our policy updates. Given suitable hyperparameters, a larger value of $\epsilon$ will enable larger updates to policy between iterates, hence larger magnitude outer gradients $\boldsymbol{g}^O$. In contrast, scaling the outer learning rate $\sigma$ does not change the size of the trust region used for $\boldsymbol{g}^O$ estimation, but instead directly scales the outer gradient when applying the update to the behavior parameters $\sigma \boldsymbol{g}^O$.

Let $\boldsymbol{\theta}^{*(\epsilon)}$ be the surrogate objective solution for a given value of $\epsilon$, defining the outer gradient $\boldsymbol{g}^{O(\epsilon)} = \boldsymbol{\theta}^{*(\epsilon)} - \boldsymbol{\theta}$. Assume the equivalence of $\epsilon$-variation and $\sigma$-variation; in other words the behavior of outer learning rates can simply be achieved by varying the clipping $\epsilon$. Formally, this implies that there exists $k \in \mathbb{R}^+$

$$\sigma \boldsymbol{g}^{O(\epsilon_0)} = \sigma(\boldsymbol{\theta}^{*(\epsilon_0)} - \boldsymbol{\theta}) = \boldsymbol{\theta}^{*(k\epsilon_0)} - \boldsymbol{\theta} = \boldsymbol{g}^{O(k\epsilon_0)},$$

for any given value of $\sigma \in \mathbb{R}^+$.

Consider a true proximal objective, such as PPO-KL (Hsu et al., 2020). Assuming the penalty coefficient $\beta$ is sufficiently high such that the proximal objective is convex, and that the inner-loop optimization converges to the global minimum thereof, the above condition is met. In this case the outer gradient corresponds to exactly the natural policy gradient (Kakade, 2001). Scaling $\beta$ directly scales the trust region size, and the (unique) solution remains on the span of the natural policy gradient. However, PPO-clip is *not* a true proximal objective, and importantly non-convex. Unlike in PPO-KL there is no guarantee of a unique solution, indeed the clipping mechanism implies regions of equivalent loss. Given that is is possible to irreversibly enter the clipped region (Engstrom et al., 2020) we cannot assume different values of $\epsilon = k\epsilon_0$ will converge onto the span of $\boldsymbol{\theta}^{*(\epsilon_0)} - \boldsymbol{\theta}$, hence do not meet the above condition for the equivalence of $\epsilon$-variation and $\sigma$-variation. An illustrative diagram of this behavior is provided in figure 25.

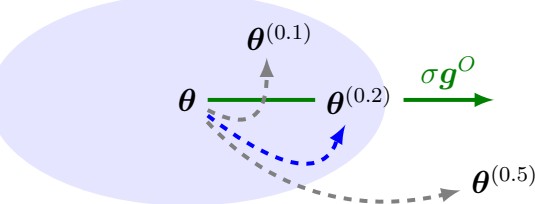

Figure 25: **The behavior permitted by varying the outer learning rate is not directly recovered by varying the clipping** $\epsilon$. Different values of clipping parameter $\epsilon \in \{0.1, 0.2, 0.5\}$ lead to different surrogate objective solutions $\boldsymbol{\theta}^{*(\epsilon)}$. Solving the surrogate objective with $\epsilon = 0.2$ (blue dashed) results in outer gradient $\boldsymbol{g}^{O(0.2)} = \boldsymbol{\theta}^{*(0.2)} - \boldsymbol{\theta}$. By varying the outer learning rate $\sigma$ we can update the parameters to any point on this vector span (green). Varying the clipping epsilon to $0.1$ or $0.5$ increases or decreases the trust region size, but this does not imply the inner loop will converge to the span of $\boldsymbol{\theta}^{*(0.2)} - \boldsymbol{\theta}$, and therefore we are unable to directly recover the outer learning rate behavior.

### G.2    INNER LEARNING RATES

The inner learning rate $\eta$ defines the learning rate for the inner-loop optimization of the clipped surrogate objective. Unlike clipping $\epsilon$, $\eta$ does not change the size of the trust region established, but instead influences the convergence to solution within this trust region. In contrast, the outer

learning rate $\sigma$ defines a rescaling of the outer gradient $\sigma \boldsymbol{g}^O$ when it is applied to update the behavior parameters.

Let $\boldsymbol{\theta}^{*(\eta)}$ be the surrogate objective solution for a given value of $\eta$, defining the outer gradient $\boldsymbol{g}^{O(\eta)} = \boldsymbol{\theta}^{*(\eta)} - \boldsymbol{\theta}$. Assume the equivalence of $\eta$-variation and $\sigma$-variation; in other words the behavior of outer learning rates can simply be achieved by varying the inner learning rate. Formally, this implies that there exists $l \in \mathbb{R}^+$

$$\sigma \boldsymbol{g}^{O(\eta_0)} = \sigma(\boldsymbol{\theta}^{*(\eta_0)} - \boldsymbol{\theta}) = \boldsymbol{\theta}^{*(l\eta_0)} - \boldsymbol{\theta} = \boldsymbol{g}^{O(l\eta_0)},$$

for any given value of $\sigma \in \mathbb{R}^+$.

Unlike clipping $\epsilon$ the inner learning rate $\eta$ does not affect the size of the trust region defined by the surrogate objective. However, as previously discussed the PPO-clip surrogate objective is non-convex and has no guarantee of a unique solution. Furthermore, it is possible to irreversibly escape the unclipped trust region. The convergence of the inner-loop to $\boldsymbol{\theta}^{*(\eta)}$ is simply defined using a specified number of inner-loop iterations. It is therefore trivial to see that variation in $\eta$ may lead to solutions not on the span of $\boldsymbol{\theta}^{*(\eta_0)} - \boldsymbol{\theta}$, hence do not meet the above condition for the equivalence of $\eta$-variation and $\sigma$-variation.

## H    COMPUTATIONAL COMPLEXITY

All outer-PPO algorithms proposed have negligible increase in computational complexity over PPO, only requiring a few vector scaling and addition operations. Outer-PPO configurations that maintain first moment estimates such as momentum-PPO and biased initialization have linear increase in memory complexity over standard PPO with respect to the parameter count. Since outer-PPO is a lightweight modification to the outer loop of vanilla PPO, it does not increase complexity with respect to dataset size (either in terms of total timesteps, or timesteps per iteration).

In table 6 we report the runtime for the four different algorithms evaluated. We use the final 64-seed evaluation runs to compute the runtime, hence hyperparameters relevant to runtime such as parallel environments etc. are fixed. We use v4-8 for all of the experiments in this table. These times are for 4-seeds to be evaluated, using our parallel implementation that distributes each seed to a different TPU device. We observe no significant deviation in runtime between the algorithms, supporting the claim of no material increase in complexity.

Table 6: Performance metrics (runtime in minutes) for each method across tasks with standard deviations. Results reported from the 64-seed evaluation.

| Task | Runtime (minutes) | | | |
|---|---|---|---|---|
| | **Baseline** | **Outer-LR** | **Outer-Nesterov** | **Biased Initialization** |
| ant | $8.6 \pm 0.5$ | $8.5 \pm 0.5$ | $8.4 \pm 0.5$ | $8.4 \pm 0.5$ |
| halfcheetah | $21.6 \pm 0.5$ | $21.6 \pm 0.5$ | $21.5 \pm 0.5$ | $21.5 \pm 0.5$ |
| hopper | $12.6 \pm 0.5$ | $12.7 \pm 0.5$ | $12.6 \pm 0.5$ | $12.7 \pm 0.5$ |
| humanoid | $16.5 \pm 0.5$ | $11.8 \pm 0.4$ | $11.9 \pm 0.3$ | $11.7 \pm 0.5$ |
| humanoidstandup | $24.2 \pm 0.4$ | $24.1 \pm 0.4$ | $24.1 \pm 0.4$ | $24.4 \pm 0.5$ |
| walker2d | $5.9 \pm 0.4$ | $5.8 \pm 0.4$ | $6.1 \pm 0.4$ | $6.0 \pm 0.4$ |
| game_2048 | $5.7 \pm 0.5$ | $5.9 \pm 0.3$ | $5.9 \pm 0.3$ | $6.0 \pm 0.4$ |
| maze | $22.3 \pm 0.5$ | $22.3 \pm 0.5$ | $22.4 \pm 0.5$ | $22.1 \pm 0.3$ |
| rubiks_cube | $5.2 \pm 0.4$ | $6.2 \pm 0.5$ | $5.5 \pm 0.5$ | $5.5 \pm 0.5$ |
| snake | $4.1 \pm 0.4$ | $4.2 \pm 0.4$ | $4.1 \pm 0.4$ | $4.1 \pm 0.4$ |
| asterix | $4.9 \pm 0.3$ | $4.0 \pm 0.0$ | $4.2 \pm 0.4$ | $4.1 \pm 0.3$ |
| breakout | $6.7 \pm 0.5$ | $6.7 \pm 0.5$ | $6.9 \pm 0.5$ | $6.8 \pm 0.4$ |
| freeway | $6.1 \pm 0.3$ | $6.3 \pm 0.5$ | $6.2 \pm 0.4$ | $6.2 \pm 0.4$ |
| space_invaders | $3.3 \pm 0.5$ | $3.1 \pm 0.3$ | $3.1 \pm 0.4$ | $3.1 \pm 0.4$ |

## I   EVALUATION DETAILS

Algorithms are evaluated across $M$ tasks within a defined suite, with $N = 64$ independent seeded runs conducted per task. Each run involves training over $1 \times 10^7$ timesteps, with performance measured over 128 episodes at 20 equally spaced checkpoints $i$. The checkpointed model achieving the highest mean return $G^i_{m,n}$ across all intermediate evaluations is selected for final 'absolute' evaluation.

In absolute evaluation, the selected model is tested over 1280 episodes to obtain more reliable performance estimates. For consistency across tasks, raw scores $x_{m,n}$ are normalized for each task $m$ and seed $n$, using the observed minimum and maximum scores found during the entire experimentation process for each task independently as proxies for the global min and max returns. This normalization produces a matrix of normalized scores $x_{1:M,1:N}$, which are aggregated to derive performance metrics.

To provide robust statistical estimates, we compute 95% confidence intervals through stratified bootstrapping over the $M \times N$ experiments. This approach accounts for variability across tasks and runs, ensuring results reflect the algorithm's performance across the full task suite.

### I.1   METRICS

We evaluate algorithm performance using the following metrics:

1. **Mean and Median Scores:** These traditional metrics summarize overall performance, with the mean capturing the average performance across runs and the median offering robustness to extreme values.

2. **Interquartile Mean (IQM):** IQM calculates the mean of the central 50% of runs, excluding the upper and lower quartiles. This metric reduces sensitivity to outliers and provides a statistically efficient estimate of performance.

3. **Probability of Improvement:** Probability of Improvement measures the likelihood that one algorithm $X$ outperforms another $Y$ on a random task $m$. It is defined using the Mann-Whitney U-statistic (Mann & Whitney, 1947) as:

$$Pr(X > Y) = \frac{1}{M} \sum_{m=1}^{M} Pr(X_m > Y_m),$$

   where:

$$Pr(X_m > Y_m) = \frac{1}{NK} \sum_{i=1}^{N} \sum_{j=1}^{K} S(x_{m,i}, y_{m,j}),$$

   and $S(x, y)$ is given by:

$$S(x, y) = \begin{cases} 1 & \text{if } y < x \\ 0.5 & \text{if } y = x \\ 0 & \text{if } y > x \end{cases}.$$

4. **Performance Profiles:** Performance profiles visually compare algorithms by plotting the fraction of runs exceeding a given performance threshold. These plots highlight stochastic dominance and performance variability.

5. **Sample Efficiency:** Sample efficiency is assessed by plotting the interquartile mean score against the number of environment steps, showing how quickly an algorithm achieves high performance.

## J   CO-OPTIMIZATION EXPERIMENTS

In the results of figures 3, 4, 8, and 9 we use base PPO hyperparameters as identified from 500 trials of baseline tuning using the Tree Parzen estimator Watanabe (2023), and outer-PPO hyperparameters as identified using grid searching where the base PPO hyperparameters are kept frozen for the grid

search. In contrast, the baseline was tuned for 600 trials using only the Tree Parzen estimator. Whilst all methods have a total budget of at most 600 trials, there is a distinction in the tuning process, as for the final 100 trial the outer-PPO methods are directly searched over a smaller hyperparameter space (1 or 2 dimensions), whereas the baseline optimization continues over the full set of base hyperparameters (11 dimensions).

To establish if the performance increases observed can be attributed to the change in tuning procedure we conduct an additional experiment in which the outer-PPO hyperparameters are co-optimized with the base PPO hyperparameters using the Tree Parzen estimator. Given that the outer-PPO hyperparameter are a superset of the baseline PPO hyperparameters, we use the 500-trial baseline sweep as a starting point for this outer-PPO tuning, where the baseline trials are edited to represent standard PPO within the outer-PPO hyperparameter space (e.g $\sigma = 1$). We then tune the union of base PPO and outer-PPO hyperparameters for 100 trials using the Tree Parzen estimator, to match the 600 trials of baseline tuning. When selecting the optimal configuration of outer-PPO hyperparameters, we take the maximum performing trial from the final 100 trials to ensure the outer-PPO configurations do not simply represent standard PPO.

Results for outer-LR on Brax are provided in figures 26 - 29. We observe the performance increases to be comparable to those reported in the previous figures 3, 4, 8 and 9. This demonstrates the tuning procedure was not responsible for the performance increases observed, and that the outer-LR can be tuned for superior performance in a given hyperparameter tuning budget under a fair, like-for-like tuning procedure. Further results for other suites and algorithms to be added in updated versions of this manuscript.

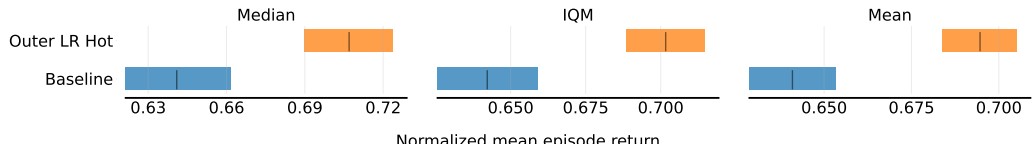

Figure 26: **Aggregate point estimates** for Brax using hyperparameters from Tree Parzen estimator co-optimization of base PPO and outer learning rate for 100 trials, using the 500-trial baseline sweep as initialization. Optimal hyperparameters *per-environment* are used. Normalized to task min/max across all experiments.

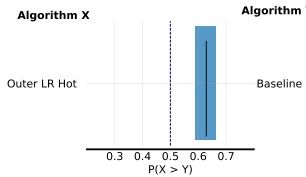

Figure 27: **Probability of improvement** for Brax using hyperparameters from Tree Parzen estimator co-optimization of base PPO and outer learning rate for 100 trials, using the 500-trial baseline sweep as initialization. Optimal hyperparameters *per-environment* are used. Normalized to task min/max across all experiments.

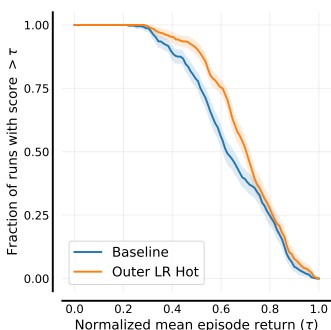

Figure 28: **Performance profiles for Brax** using hyperparameters from Tree Parzen estimator co-optimization of base PPO and outer learning rate for 100 trials, using the 500-trial baseline sweep as initialization. Normalized to task min/max across all experiments.

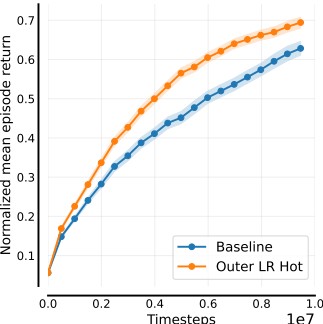

Figure 29: **Sample efficiency curves** for Brax using hyperparameters from Tree Parzen estimator co-optimization of base PPO and outer learning rate for 100 trials, using the 500-trial baseline sweep as initialization. Normalized to task min/max across all experiments.

## K    PPO HYPERPARAMETER SENSITIVITY

In figures 5 - 7 we plot the mean normalized return for the 4-seed grid searches used to select hyperparameter for final evaluation. Whilst noisier than the final 64-seed evaluation, these plots provide insight into the hyperparameter sensitivity of the outer-PPO methods.

In this appendix we conduct a corresponding analysis of standard PPO, by grid searching two hyperparameters; learning rate scale and $\epsilon$-scale. learning rate scale scales the actor and critic (inner) learning rates, and $\epsilon$-scale scales the clipping $\epsilon$. As in the outer-PPO hyperparameter sweeps, we use the optimal base PPO hyperparameters as identified up to 500 trials of baseline tuning. In figure 30 we plot the results of these grid searches.

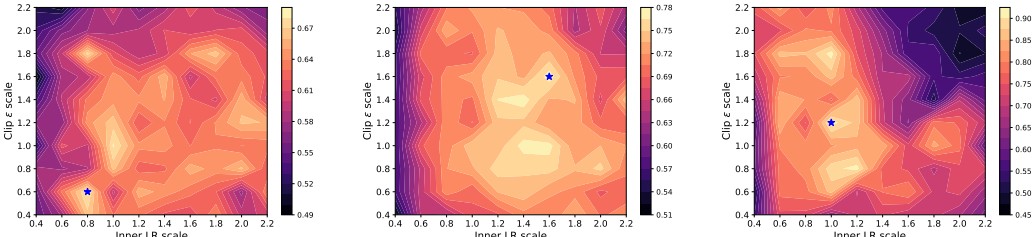

Figure 30: **Learning rate scale and epsilon scale sensitivity plots.** Mean normalized return across the Brax (left), Jumanji (center), MinAtar (right) tasks as a function of learning rate scale and $\epsilon$-scale. Mean of 4 seeds plotted. Normalized to task min/max across all experiments.

