# OpenReview forum: "Beyond the Boundaries of Proximal Policy Optimization"
_ICLR.cc/2025/Conference — Submitted to ICLR 2025_

### Official Review · Reviewer_z32D · 2024-11-01

**Soundness:** 2
**Presentation:** 2
**Contribution:** 2
**Rating:** 5
**Confidence:** 2

**Summary:**

This paper introduces outer-PPO, a novel perspective of proximal policy optimization that applies arbitrary gradient-based optimizers to the outer loop of PPO.

**Strengths:**

1. The idea of the proposed method (outer-PPO) is interesting.
2. This paper provides a comprehensive evaluation of the proposed method.

**Weaknesses:**

1. In section 5.1, it would make this paper stronger if the authors could explain metrics in detail.
2. The analysis in section 5.1 could dive deeper into why, in some environments, the methods proposed in this paper work well, while in other environments, these methods are not better than the baseline PPO. For example, the authors could provide some hypotheses and use some experiments to support these hypotheses.
3. In section 5.2, hyperparameter sensitivity, it would make this paper stronger if the authors could compare the hyperparameter sensitivity of their methods with baseline PPO.
4. Given Figure 8 and Figure 9 in the appendix, it is unclear if the final converged performance of the proposed methods would be better than baseline PPO since it seems all algorithms still haven’t converged.

Suggestions:
1. In the experiments that the authors conducted, it seems the proposed methods do not show better performance than baseline PPO, especially since it is unclear which implementation version of baseline PPO the authors used. Currently, there are many different implementations of PPO, and different implementations of PPO can significantly affect the final performance [1]. For research focus, it is acceptable that the proposed method does not perform better than PPO in general environments. However, it would make the paper stronger if the authors could provide evidence in which situations (such as environments with high dimensional input or output), the proposed method is better than PPO.
2. It would make this paper stronger if the authors could provide the details of the training time of each algorithm with the same number of timesteps.
3. Page 4, Line 195, appendix ??.

[1] The 37 Implementation Details of Proximal Policy Optimization

**Questions:**

Check the weaknesses section above.

---

> ### Author Response · Authors · 2024-11-20
> **Response to Reviewer z32D**
>
> We thank the reviewer for their valuable and constructive feedback. We are encouraged by their comment that "the idea of the proposed method (outer-PPO) is interesting” and that "this paper provides a comprehensive evaluation of the proposed method."
>
> ## Weaknesses:
>
> **"In section 5.1, it would make this paper stronger if the authors could explain metrics in detail."**
>
> For evaluation we used the RLiable library implementation of the metrics introduced in [1]. We have amended the manuscript to include a description of each metric and how it is calculated in Appendix I.
>
> **"The analysis in section 5.1 could dive deeper into why, in some environments, the methods proposed in this paper work well, while in other environments, these methods are not better than the baseline PPO. For example, the authors could provide some hypotheses and use some experiments to support these hypotheses."**
>
> Outer-PPO shows statistically significant improvements in the Brax and Jumanji  suites but not in MinAtar. We provide a detailed discussion of the potential reasons for this in lines 202 - 215. We have also added additional justifications for this as follows: (i) our MinAtar baseline results are significantly higher than other recent work [2] (ii) our strong baselines are in fact approaching the mathematical limit of performance on the MinAtar suite as defined using the gymnax library. We additionally comment that our PPO baseline is 2.5x as strong as the PPO baseline used by [2] for the same timestep budget. We respectfully believe the results on Brax and Jumanji are sufficient evidence to support our claims of superior performance of outer-PPO, and the non-optimality of the three design choices highlighted in lines 47 - 52.
>
> **"In section 5.2, hyperparameter sensitivity, it would make this paper stronger if the authors could compare the hyperparameter sensitivity of their methods with baseline PPO."**
>
> We thank the reviewer for this suggestion in improve our work. We have added figures concerning the hyperparameter sensitivity of standard PPO, due to space limitations these are provided in Appendix K.
>
> **"Given Figure 8 and Figure 9 in the appendix, it is unclear if the final converged performance of the proposed methods would be better than baseline PPO since it seems all algorithms still haven’t converged."**
>
> The reviewer is correct that our experiments were conducted with a sample budget of 1e7 transitions, which was insufficient for convergence in a variety of the tasks. This was a deliberate choice to focus on sample efficiency under constrained budgets. We additionally made this decision to enable large-scale hyperparameter tuning within a tractable computational budget, to ensure our baselines were a strong reference point against which we could measure algorithmic progress. We acknowledge this limitation explicitly in line 485 of the manuscript. Despite this limitation, we believe our results demonstrate effectively the improved sample efficiency of outer-PPO variants over baseline PPO.

---

> > ### Author Response · Authors · 2024-11-20
> >
> > ## Suggestions
> >
> > **"In the experiments that the authors conducted, it seems the proposed methods do not show better performance than baseline PPO, especially since it is unclear which implementation version of baseline PPO the authors used..."**
> >
> > We respectfully disagree that the proposed methods do not show better performance over baseline PPO. Figures 3, 4, 8, and 9 show a robust, statistically significant improvement across three metrics (median, IQM, mean) over an aggressively tuned per-task PPO baseline. We understand there may be confusion with the hyperparameter sensitivity plots, where common outer-PPO hyperparameters are shared across a suite to represent normalized performance across the grid range.  We have amended the captions to Figures 5, 6 and 7 to make this distinction more clear. We emphasize that the sensitivity plots share outer hyperparameters hence do not achieve the stronger performance of the task-specific hyperparameters used in Figures 3, 4, 8, and 9. However despite the lack of clear improvement given common hyperparameters, the sensitivity plots do show that learning is stable and performance is robust for a wide range of values, particularly on Brax and Jumanji. We lastly highlight that the inner PPO hyperparameters are themselves task-specific hence we would not expect common values to be optimal.
> >
> > We thank the reviewer for highlighting the lack of clarity concerning the PPO implementation, and share their concern for the sensitivity of PPO to implementation details. As stated in line 813 we used the Stoix implementation of PPO, which is publicly available here (https://github.com/EdanToledo/Stoix). We have also provided our code implementation in the supplementary material. To further provide clarity on the implementation details we have provided an additional Table 1 in Appendix B in which we exhaustively define the implementation details employed (or omitted) in our work as identified by the works [3, 4]. We thank the reviewer for highlighting this, as we believe this addition will enhance the reproducibility and transparency of our results.
> >
> > **"It would make this paper stronger if the authors could provide the details of the training time of each algorithm with the same number of timesteps."**
> >
> > We have added Appendix H in which we discuss the computational complexity of the outer-PPO methods, and provide a Table 6 in which the runtimes are compared. As the outer-PPO methods do not materially increase the computational complexity, we observe no significant deviation in runtime between the different methods.
> >
> > We deeply appreciate your time and effort in reviewing our work and providing valuable feedback. We hope that our response and amendments, particularly the addition of Appendices H, I and K, address your concerns and if so, we would be grateful if you could consider updating your score. We would of course of happy to respond to any remaining questions you may have.
> >
> > [1] Deep Reinforcement Learning at the Edge of the Statistical Precipice
> > [2] Discovered Policy Optimisation
> > [3] Implementation Matters in Deep Policy Gradients: A Case Study on PPO and TRPO
> > [4] The 37 Implementation Details of Proximal Policy Optimization

---

> > > ### Comment · Reviewer_z32D · 2024-11-30
> > >
> > > Thank you for your response. After carefully considering your rebuttal and the concerns raised by other reviewers, I have decided to keep my original score.

---

### Official Review · Reviewer_xVTT · 2024-11-02

**Soundness:** 2
**Presentation:** 3
**Contribution:** 1
**Rating:** 3
**Confidence:** 5

**Summary:**

The typical PPO algorithm performs an inner loop optimization at each policy update step. With a fixed trajectory data and current policy parameter $\theta$, it optimizes an update to the policy parameters $\theta’$, then applies the new parameters to get more trajectory data. The current paper proposes to modify this update rule by instead considering $\Delta = \theta’ - \theta$ as a replacement to the gradient in any typical gradient-based optimizer. For example, one can consider the update $\sigma \Delta$. In this case $\sigma=1$ will correspond to the standard PPO, but by changing $\sigma$ it generalizes the algorithm. As the original PPO is included in this class of updates, tuning the parameter is guaranteed to at least not decrease the performance compared to PPO. They compare performance on Brax continuous control tasks, MinAtar and Jumanji, and through an extensive hyperparameter tuning procedure claim 5-10% improved performance for the same number of hyperaprameter optimization trials (there were some differences in the hyperparameter optimization procedure between outer-PPO and regular PPO).

**Strengths:**

- The paper is clear.
- There do a lot of experiments, and the results are reported faithfully.

**Weaknesses:**

- The performance improvements are at best small (5-10%)
- The performance change could also be due to changes in hyperparameter tuning procedures.
Currently, the authors tune PPO by optimizing 11 hyperparameters for 600 trials (each trial averaged across 4 seeds) using a Tree Structured Parzen method, and then doing a final evaluation at the best parameters for 64 seeds (different from the initial 4 seeds). And the outer-PPO methods are hyperparameter tuned by first doing 500 trials of PPO hyperparameter optimization, then doing an additional 100 trials by a hyperparameter sweep over the outer-PPO parameters. These two optimization methods are inconsistent, as outer-PPO includes a direct sweep over the hyperparameters. Such an inconsistency in tuning may lead to an improved performance for outer-PPO. There is no guarantee that, for example, the standard PPO may also not perform better if it first does 500 trials of optimization with the Tree Structured Parzen, and then does a 100 trial sweep over some chosen important hyperparameters. In general, while I appreciate that the authors tried to tune the parameters exhaustively, 11 hyperparameters are a lot to tune, and with the evaluation noise due to using only 4 seeds it is difficult to create a fully convincing evaluation method where it would be clear that small improvements like 5-10% are due to an improvement in the algorithm, and not some small details in the tuning procedure. Perhaps a simpler experimental protocol would be more convincing. For example, tune PPO  on all tasks, then employ a single fixed outer learning rate (the same one across all tasks). If such a simple procedure lead to an improved performance, it would be more convincing.

- The optimal outer-PPO hyperparameters are different per task.
- No strong justification is given why we should expect the new method to lead to large improvements in performance (whilst it is clear that the performance should not decrease under proper tuning, there is no indication that large improvements are expected.)
- The outer learning rate sensitivity plots show that the peak performance is not much different compared to $\sigma=1$, which corresponds to standard PPO

**Questions:**

For the outer-PPO hyperparameter tuning, did you also first tune using 4 seeds, and then run a final evaluation using 64 new seeds, just like for the standard PPO experiments?

In general, I did not find the paper interesting, and I would expect much larger improvements in performance (2x improvement or something around that) for me to recommend the work for publication. There is no strong justification for why the proposed method solves a fundamental limitation in PPO or why it would lead to a fundamental improvement in performance. I do not expect any practitioner to adopt the proposed algorithm. For this kind of work, I would expect large empirical performance gains for me to recommend it for publication. Therefore, I do not foresee myself changing my score unless such results are provided in the rebuttal. I would recommend aiming to submit the work to a venue that de-emphasizes significance, and merely looks at correctness of the claims.

Typos:
“Given it’s” → “Given its”
Line 195: “appendix ??” the reference link is missing.

---

> ### Author Response · Authors · 2024-11-20
> **Response to Reviewer xVTT**
>
> We thank the reviewer for their comments that “the paper is clear”, and that we “do a lot of experiments, and the results are reported faithfully.”
>
> **“The performance improvements are at best small (5-10%)”**
>
> The reviewer later comments they would expect a **“much larger improvement in performance”** on the order of **“2x improvement”**.
>
> As stated in line 99, we acknowledge the improvements are in the range of (5 - 10%). However, we believe the this improvement must be considered in light of our extensive baseline tuning (600 4-seeds trials), number of seeds evaluated (64), and statistical significance of these results (as evidenced by error bars in Figure 3). Whilst some works may claim a larger improvement, we instead focused on reporting robust algorithmic improvement by ensuring we have a strong baseline. We believe our evaluation procedure to be notably stringent and tranparent compared to other works, with other works accepted to top venues often employing 5 - 10 evaluation seeds and providing limited details on hyperparameter tuning for both baselines and proposed methods. We additionally comment that even given suboptimal baselines and fewer evaluation seeds, many comparable works accepted to top venues do not achieve the 2x improvement as requested by the reviewer. To further highlight the challenges in identifying algorithmic progress over suboptimal baselines, we compare our baseline PPO results on MinAtar to those reported by Discovered Policy Optimisation [1], using the same gymnax implementation and timestep budget of 1e7. Comparing our results to theirs, we see our PPO baseline is a 2.5x improvement over the PPO baseline of [1].
>
> **“The performance change could also be due to changes in hyperparameter tuning procedures…”**
>
> We thank the reviewer for highlighting this inconsistency in our hyperparameter tuning methodology. To address this we are running an additional experiment in which we tune outer-PPO methods using the Tree Parzen estimator. Given that the outer-PPO hyperparameters are superset of the baseline PPO hyperparameters, we use the 500-trial baseline sweep as a starting point for this outer-PPO tuning. We then tune the union of base PPO and outer-PPO hyperparameters for a further 100 trials giving each algorithm the exact same total number of trials (600) using the exact same algorithm (Tree Parzen estimator) as the original baseline results. We believe this should resolve the reviewers' concerns in terms of differing tuning procedures, as the outer-PPO tuning problems are now more challenging (12 / 13 hyperparameters) compared to the baseline tuning (11 hyperparameters) from the 500-trial point at which they diverge. With these results, we will be able to accurately compare the algorithms within a fixed hyperparameter tuning budget for the same methodology. We have added experimental details and preliminary results on this new evaluation process in Appendix J. The preliminary results demonstrate improved performance of outer-LR on Brax over the baseline, of comparable magnitude to the grid-searched results. This new evaluation methodology has the additional advantage of addressessing a previous limitation of our work, “lack of co-optimization” so we sincerely thank the reviewer for highlighting this issue. We will update the paper and respond with updated results when have obtained them.
>
> **"For example, tune PPO on all tasks, then employ a single fixed outer learning rate (the same one across all tasks). If such a simple procedure lead to an improved performance, it would be more convincing."**
>
> We unfortunately do not have the capacity at this stage to rerun the baseline tuning using common values across suites, but appreciate the reviewers suggestion for improvements to the work. We hope the results of our new outer-PPO tuning will appease their concerns on this matter.
>
> **"The optimal outer-PPO hyperparameters are different per task"**
>
> We appreciate the reviewers comment but respectfully highlight that the base PPO hyperparameters are different for each task, hence it can be expected that a method building on this base configuration may require per-task tuning for optimal performance. These base hyperparameters affect the surrogate optimization process (e.g the size of the trust region via $\epsilon$) which in turn will affect which outer PPO hyperparameters will be optimal.

---

> > ### Author Response · Authors · 2024-11-20
> >
> > **"No strong justification is given why we should expect the new method to lead to large improvements in performance"**
> >
> > We wish to draw the reviewers attention to the three research questions we highlight in lines 48 - 52. The contribution of this work is to (a) identify these implicit design choices exist, (b) propose a method than enables these design choices to be relaxed, and \(c\) empirically validate that in all three cases the design choice is suboptimal. We believe the common understanding before our work would have been that all three design choices are necessary for optimal performance, particularly the outer learning rate of 1 and lack of outer loop momentum. Therefore, our results stand in contrast to the field's understanding of one of the most commonly used algorithms across reinforcement learning, which we believe empirically alone is a strong research contribution. We further provide intuition for how these methods are modulating the PPO update rule, and how these can be motivated in Sections 3.1, 3.2 and 3.3. We appreciate the reviewers comment that a stronger justification would be welcomed, but believe this is beyond the scope of this initial empirical investigation, which we hope will stimulate further research both empirical and theoretical to understand the properties of outer-PPO and outer-variants of other policy gradient methods.
> >
> > **"The outer learning rate sensitivity plots show that the peak performance is not much different compared to sigma=1, which corresponds to standard PPO"**
> >
> > We understand there may be confusion with the hyperparameter sensitivity plots, where common outer-PPO hyperparameters are shared across a suite to represent normalized performance across the grid range. We have amended the captions of Figures 5, 6 and 7 make this distinction more clear.  We emphasize that the sensitivity plots share outer hyperparameters hence do not achieve the stronger performance of the task-specific hyperparameters used in Figures 3, 4, 8, and 9. However despite the lack of clear improvement given common hyperparameters, the sensitivity plots do show that learning is stable and performance is robust for a wide range of values, particularly on Brax and Jumanji. We lastly highlight that the inner PPO hyperparameters are themselves task-specific hence we would not expect common values to be optimal.
> >
> > **"For the outer-PPO hyperparameter tuning, did you also first tune using 4 seeds, and then run a final evaluation using 64 new seeds, just like for the standard PPO experiments?"**
> >
> > Yes, the grid search was performed using 4 seeds per trial with final evaluation using 64 new seeds. The same set of seeds was used for evaluation on all methods.
> >
> > We are sincerely thankful for your thoughtful review and the time you have taken to engage with our work. Should our explanations and amendments sufficiently address your concerns, we kindly ask you to consider your score. We would be eager to engage further if you have any questions remaining.

---

> > > ### Comment · Reviewer_xVTT · 2024-11-24
> > >
> > > Thank you for the response. I am still not convinced regarding the points below:
> > >
> > > >“The performance improvements are at best small (5-10%)”
> > > The reviewer later comments they would expect a “much larger improvement in performance” on the order of “2x improvement”.
> > >
> > > >“The performance change could also be due to changes in hyperparameter tuning procedures…”
> > >
> > > >"No strong justification is given why we should expect the new method to lead to large improvements in performance"
> > >
> > > > "The outer learning rate sensitivity plots show that the peak performance is not much different compared to sigma=1, which corresponds to standard PPO"
> > >
> > > It is always possible to create arbitrary variations of algorithms that include the initial algorithm as a special case. In such cases, we would always expect that performance can only improve compared to the original algorithm if we tune the hyperparameters.
> > > Therefore, I think it is important to have justification for why we would expect the new proposal to either be crucial in some special cases, or why we might expect it to lead to significant performance improvements. The current method does not have such theoretical justification. While in principle smaller improvements than 2x are also fine, for the current work, without strong justifications why the method is good, I would expect larger empirical performance improvements to justify me voting for accept (the better performance improvement could be shown either through larger gains, or through more consistent results, e.g., by using shared hyperparameters, but the current results are not convincing to me. I don't think anyone would bother with the proposal, as it is necessary to tune task-wise parameters, and the gains are still at best small).

---

### Official Review · Reviewer_LKWg · 2024-11-03

**Soundness:** 3
**Presentation:** 3
**Contribution:** 2
**Rating:** 6
**Confidence:** 4

**Summary:**

This paper proposes an improvement of Proximal Policy Optimization (PPO), one of the most well-known online policy gradient algorithms.

PPO works by following two nested optimization stages: the inner optimization optimizes a "clipped surrogate objective" that can be seen as a constrained optimization problem, where the optimal solution optimizes the expected advantage while keeping the target policy close to the data; the outer optimization stage simply loops simply updates the target policy and collects new data.

In PPO, the inner objective is optimized by gradient ascent, while the outer objective is seen simply as a loop.

The authors' idea is to see the inner optimization as a gradient estimation procedure and the outer loop as a gradient ascent procedure. By leveraging this idea, the authors explore different learning rates (which in PPO is inherently set to 1), the application of momentum, and the possibility of "biasing" the initialization of the inner loop by leveraging on the outer gradient ascent.

**Strengths:**

Originality
--------------

The idea presented in the paper is, as far as I know, novel.

Quality and Clarity
-------------------------

The paper presents the idea in a clear way, highlighting the main research questions well and providing a robust empirical analysis of the proposed algorithm (with very detailed ablation studies). The algorithm is sound and coherent with the investigation objective.

Significance
----------------

I am unsure about the significance of the proposed idea. The paper introduces new hyperparameters that make PPO more complicated rather than more simple. I am unsure whether the complexity introduced is a payoff of the little gains in terms of performance. However, the takeaway message of seeing the outer loop of PPO as a gradient ascent procedure is interesting.

**Weaknesses:**

As I have mentioned above, I think a weakness of the proposed method is the introduction of new hyperparameters (i.e., outer learning rate and momentum) - with what seems to be little payoff.

Furthermore, I am unsure whether the learning rate is necessary: the hyperparameter $\epsilon$ of PPO already provides a mechanism to control "how aggressive" the policy updates are. While I can acknowledge that the learning rate and the $\epsilon$ are two different terms (i.e., the learning rage $\sigma$ acts in the parameter space, while the hyperparameter $\epsilon$ acts in the "policy" space), I can't see what is the advantage of using $\sigma$ in place of modifying $\epsilon$.

Perhaps, as highlighted in the "Strength" section, the main weak point of the paper is the significance.

**Questions:**

I ask the authors to clarify what is the advantage of introducing the learning rate instead of modifying $\epsilon$.

---

> ### Author Response · Authors · 2024-11-20
> **Response to Reviewer LKWg**
>
> We thank the reviewer for their thoughtful feedback, highlighting that the work is "novel", and presented in a "clear way". We additionally thank the reviewer for recognising that we provide a "robust empirical analysis of the proposed algorithm (with very detailed ablation studies)" and that "the algorithm is sound and coherent with the investigation objective". We lastly thank the reviewer for stating that “the takeaway message of seeing the outer loop of PPO as a gradient ascent procedure is interesting.”
>
> **“The paper introduces new hyperparameters that make PPO more complicated rather than more simple. I am unsure whether the complexity introduced is a payoff of the little gains in terms of performance.”**
>
> A related comment of the reviewer follows:
>
> **“I think a weakness of the proposed method is the introduction of new hyperparameters (i.e., outer learning rate and momentum) - with what seems to be little payoff”**
>
> We agree that adding new hyperparameters increases complexity, but we emphasize that the practical implementation of our method involves minimal changes to the PPO framework - approximately a five-line code modification. We further emphasize the best performing method (outer learning rates) introduces only a single extra hyperparameter. We also highlight that our evaluation gave all methods (both baseline PPO and outer-PPO methods) the same hyperparameter tuning budget, hence outer-PPO is in fact easier to tune for higher performance than standard PPO. Additionally, the hyperparameters introduced (outer learning rate and momentum) were shown to be, on average, robust across a range of values, as demonstrated in Figures 3 and 4. Lastly, the performance gains achieved are non-trivial when considering the strength of the PPO baseline. Our baseline was aggressively tuned using extensive sweeps (600 4-seed trials per task), achieving higher performance than the proposed methods of other works, and yet, outer-PPO achieved consistent improvements (5–10%) in Brax and Jumanji environments. These results demonstrate that even with strong baselines, our method yields statistically significant gains, which validates its practical utility.

---

> > ### Author Response · Authors · 2024-11-20
> >
> > ## Response to Questions:
> >
> > **“I ask the authors to clarify what is the advantage of introducing the learning rate instead of modifying $\epsilon$.”**
> >
> > The reviewer also provided further context on this question in the “Weaknesses” section of their review.
> >
> > **“Furthermore, I am unsure whether the learning rate is necessary: the hyperparameter of PPO already provides a mechanism to control "how aggressive" the policy updates are. While I can acknowledge that the learning rate and the $\epsilon$ are two different terms (i.e., the learning rage acts in the parameter space, while the hyperparameter acts in the "policy" space), I can't see what is the advantage of using $\sigma$ in place of modifying $\epsilon$.”**
> >
> > As the reviewer has noted, $\epsilon$ controls the size of the trust region in policy space, while the outer learning rate operates in parameter space, scaling the outer gradient vector directly. This decoupling allows outer-PPO to amplify or attenuate updates without altering the surrogate objective. We have added an appendix highlighting the new behavior introduced by outer-PPO, that cannot be recovered by standard PPO in Appendix G. We motivate these new behaviors in lines 202 - 215. Attenuating the update direction can be motivated as ‘not trusting’ any given outer gradient, for reasons such as the noise present in data collection and stochastic optimization, irrespective of the outer gradient magnitude (e.g using different values of $\epsilon$). Another motivation is the potential lack of monotonicity of improvement along the linear interpolation $\theta_k + \sigma(\theta_k^* - \theta_k)$ for $\sigma \in[0,1]$ arising from the non-linear map from parameters to policy and non-convex surrogate objective, which we have added in line 207. Amplifying the update vector with an outer learning rate $\sigma > 1$ can be motivated as encoding confidence in the update direction. This may be desirable to use on well-estimated low-$\epsilon$ outer gradients. We posit in the paper that optimizers in the outer loop (e.g. momentum) enhance performance in ways that modifying epsilon alone cannot achieve, as evidenced by the superior performance within a given hyperparameters tuning budget. We lastly draw the reviewers attention to the three research questions we seek to resolve in the introduction, and emphasize that we did not seek to find the highest performing outer-PPO configuration, but to answer the aforementioned questions regarding implicit design choices of standard PPO.
> >
> > **“Perhaps, as highlighted in the "Strength" section, the main weak point of the paper is the significance”**
> >
> > We view this work as a proof-of-concept exploration into the outer-loop optimization using PPO surrogate optimization as the inner-loop. The significance lies not only in the observed performance gains but also in challenging previously unquestioned assumptions in PPO design. For example, the success of outer learning rates $\sigma > 1$ is surprising given PPO's original motivation of conservative policy updates. This finding suggests opportunities for further innovation in dual-loop algorithms, as our approach is broadly applicable beyond PPO to other RL algorithms. We also underscore the rigor of our investigation, including robust baseline comparisons and comprehensive hyperparameter sensitivity analyses. This methodological contribution supports future research in outer-loop methods for reinforcement learning.
> >
> > Thank you for your thorough review and the effort you have invested in helping us improve our work. We hope our comments and the addition of Appendix G address your concerns and if so, would greatly appreciate your consideration in revising your score. We are of course eager to answer any further questions you may have.

---

> > > ### Comment · Reviewer_LKWg · 2024-11-26
> > > **Thanks**
> > >
> > > Dear authors, thanks for the time you invested in answering my review. I read carefully your rebuttal, and the modifications apported to the manuscript. I've particularly focused on Appendix G.  I don't have further questions to ask.

---

### Official Review · Reviewer_xV9r · 2024-11-04

**Soundness:** 3
**Presentation:** 3
**Contribution:** 1
**Rating:** 3
**Confidence:** 2

**Summary:**

The paper designs a novel framework, called outer-PPO, to further modify PPO’s trust region gradients through an outer loop. PPO conducts several gradient updates using each set of collected data. The proposed framework computes these gradients in the inner loop without updating and combines these gradients into one update with extra stepsize and momentum designs in the outer loop. The proposed algorithms are evaluated on Brax, Jumaji, and MinAtar environments, showing statistically significant improvement on Brax and Jumaji and comparable performance on MinAtar.

**Strengths:**

The algorithms are clearly introduced with the help of figure representations. The paper makes claims on its empirical performances, which are well supported by the empirical results.

**Weaknesses:**

The novelty of the algorithm can be explained more. Empirically speaking, outer-PPO with a non-unity learning rate performs best on Brax and Jumaji and functions as the main contribution. However, what is the difference between this proposed algorithm and rescaling the learning rate of the original PPO? Is the empirical result suggesting that stochastic gradient descent can be a better optimizer than the commonly used Adam?

**Questions:**

1. Line 37: why is “exactly coupled” emphasized? Some newly defined terms, like the behaviour parameters, can be emphasized instead.

2. What is the intuition behind the biased initialization? Trust region is used to reduce the off-policy distribution shift, but biased initialization worsens the situation.

3. Why were Brax and Jumanji chosen instead of MuJoCo or the DeepMind suite?

4. Line 303, during the hyperparameter tuning, does each trial represent a choice of the hyperparameters, and does each agent represent a random seed? Can learning curves for the baseline be provided? How is the final performance of the baseline? Could you help me read Figure 10? What are the points and meanings of the x-axis and y-axis?

5. In Figure 3, how is the metric, optimality gap, defined?

6. The result in Figure 4 is very straightforward. How is this probability metric computed? From Figure 3, there is no significant difference between algorithms. However, the probability of the proposed algorithms being better than the baseline is larger than 0.5. What causes the probability measure to lean towards proposed algorithms?

7. Line 313, “normalizing with the min/max return found for each task across all trained agents (including sweep agents).” What are sweep agents? How is the normalization conducted with the min/max return? In line 470, why does a 0.9 peak normalized return imply less variance compared to 0.7 or 0.8?

8. For the outer loop, is the gradient update applied without using any optimizers?

9. Are these outer loop hyperparameters task-dependent? Could they differ a lot for each task?

10. What does a smaller stepsize on a policy improvement direction suggest? One hypothesis can be the PPO’s gradient direction is not toward policy improvement. However, the algorithm won’t learn in this case, and this hypothesis does not seem to hold.

---

> ### Author Response · Authors · 2024-11-20
> **Response to Reviewer xV9r**
>
> We thank the reviewer for their thoughtful evaluation of our paper. We particularly thank the reviewer for their comment that the “algorithms are clearly introduced”, and that our claim of improved empirical performance is “well supported by the empirical results”.
>
> **“The novelty of the algorithm can be explained more”**
>
> Our paper introduces a novel perspective to policy gradient methods, such as PPO, in which the estimation and application of the update direction is decoupled. This decoupling highlights the distinction between the inner-loop trajectory in solving a given surrogate objective, and the outer-loop trajectory given by the sequence of behavior parameters. This insight enables us to apply arbitrary gradient-based optimizers to the outer loop of PPO. Using optimizers other than unity-learning rate gradient ascent (which corresponds to standard PPO), enables the iterative sequence of behavior parameters to be exploited.
>
> **“What is the difference between this proposed algorithm and rescaling the learning rate of the original PPO?”**
>
> We understand the confusion regarding how our proposed algorithm applying outer learning rates $\sigma$, differs from simply rescaling the (inner) learning rate $\eta$ in PPO. To help understand the distinction, we would like to draw the reviewers attention to Algorithm 2 and Algorithm 3. Scaling $\eta$ would take place in line 2 (PPOIteration, as defined in Algorithm 6) of both algorithms, whereas $\sigma$ is applied in line 3 of Algorithm 3. To further clarify, we added Appendix G containing an explanation of the non-equivalence of scaling $\sigma$ and $\eta$. To give a brief explanation, $\eta$ affects the solution of the surrogate objective, hence also the outer gradient. In contrast $\sigma$ does not affect the surrogate objective solution, but simply scales the outer gradient when it is applied as an update.
>
> **“Is the empirical result suggesting that stochastic gradient descent can be a better optimizer than the commonly used Adam?”**
>
> We apologize for any confusion on this matter. Our empirical results do not suggest that stochastic gradient descent is superior to Adam. In fact, we used Adam for the inner optimization loop of our algorithm, as in the vast majority of PPO implementations. We have amended our work to make this more clear in Appendix A.1, where we define the inner optimization loop of PPO.

---

> > ### Author Response · Authors · 2024-11-20
> >
> > ## Response to Questions
> >
> > 1. We have used "exactly coupled" to emphasize a property of standard PPO, where the behavior parameters at each iteration are exactly the solution to the previous inner-loop surrogate objective, without any additional transformations. While the term "behavior parameters" is not new in reinforcement learning (referring to the parameters defining the data-collecting policy), we agree that emphasizing this term would improve clarity, and have amended this line to do so.
> >
> > 2. The intuition behind biased initialization is to leverage prior trajectory information to improve the inner-loop optimization. One of the hypotheses of this work was that this trajectory contained useful information for solving the surrogate objective. To explore this hypothesis we apply a momentum-based step to the parameters before starting the inner-loop optimization, aiming to provide a better initialization for the inner loop optimization. Here, by ‘better initialization’ we mean an initialization closer to a solution of the surrogate objective, hence easier to optimize for. The reviewer's concern about off-policy distribution shift is valid, although, using the biased initialization does not affect the data collection nor the establishment of the surrogate objective / trust region. The trust region size (defined by $\epsilon$) is the principal component of PPO that moderates off-policy distribution shift. As biased initialization does not affect the trust region size, it should not directly increase the off-policy distribution shift.
> >
> > 3. Brax and Jumanji were chosen because they are implemented in JAX, which allows for high-performance parallel computation and end-to-end compilation of RL training. This enabled us to perform extensive experimentation and hyperparameter sweeps within a reasonable computational budget. Furthermore, these simulators offer sufficiently diverse tasks, making them well-suited for evaluating the generality of our approach. Whilst there are more extensive suites, we felt the ability to tune a strong baseline against which we could reliably measure progress took precedence.
> >
> > 4. Yes, each trial represents a distinct choice of hyperparameters, and each agent represents a random seed. The performance of each trial is averaged over four agents (i.e., four random seeds) trained using the same hyperparameters. Learning curves for the baseline w.r.t environment timesteps are provided in Figure 9, with individual task learning curves in Figures 22 - 24; the final performance of the baseline can be determined from these plots. Figure 10 shows the performance during baseline tuning using the Tree Parzen estimator; the x-axis represents the trial number and the y-axis represents the mean return of the 4 agents trained for that trial. In this figure we observe the best performing trial (highest point on y-axis) increases as more trials are completed (moving right on the x-axis), albeit with diminishing improvements for many tasks. We have amended the caption to make this clearer for the reader, and added a red line showing the highest performing trial thus far.
> >
> > 5. The optimality gap is a metric provided by the RLiable library (https://github.com/google-research/rliable). Using the notation of their work, optimality gap is defined as $\gamma − mean$, where $\gamma$ is a defined value of ‘optimality’. Given that we normalize our return by the minimum and maximum  achieved across all agents during the sweeps, and set $\gamma = 1$, the optimality gap plot simply mirrors the mean plot. Since this metric is redundant in our setting, we have removed it from the manuscript.

---

> > > ### Author Response · Authors · 2024-11-20
> > >
> > > 6. This probability metric follows the methodology in [1]. It measures the likelihood that one algorithm outperforms another on a randomly selected task, irrespective of the size of improvement. While Figure 3 shows the magnitude of performance differences, the probability metric in Figure 4 focuses solely on the robustness of improvements across tasks. The larger probability for the proposed algorithms reflects their consistent outperformance, albeit with potentially small gains. We have amended the paper to outline how this metric is calculated in Appendix I.
> > >
> > > 7. "Sweep agents" refers to all agents trained during baseline hyperparameter sweeps and the outer-PPO grid searches. The normalization uses the minimum and maximum return achieved across all experiments (including evaluation agents), the values of which are stated in Table 5. Practically this means the min and max absolute mean episode return ever observed in our entire experimentation process was used as proxies for the global min and max returns of the tasks. The formula for normalization is
> > > $$x_{\text{normalized}} = \frac{x - \text{min}(X)}{\text{max}(X) - \text{min}(X)}$$
> > > 	The performance on each task is normalized separately with their min and max values before aggregation. In line 470, we comment that the peak normalized return of 0.9 observed in Figures 5 / 6 / 7 indicates less variance in the optimal performance of MinAtar compared to Brax and Jumanji which have a peak normalized returns around 0.7 and 0.8 respectively. To understand this comment, recall that the plots show the mean of 4 seeds normalized to the maximum and minimum values found on this task. That on MinAtar it is was possible to achieve 0.9 mean normalized return in each of the three sweeps (outer-LR, outer-Nesterov, biased initialization) indicates it is possible to reliably achieve performance approaching the maximum value. In contrast on Brax and Jumanji we only achieve 0.7 and 0.8 normalized return, indicating the highest performing agent observed had significantly higher return achieved than any 4-seed trial in the grid. From this we infer that there is less variance in the optimal performance on MinAtar, as we are able to reliably achieve 0.9 mean normalized return.
> > >
> > > 8. For outer-LR the outer gradient update is applied using the standard SGD update equation (Algorithm 3, line 3). For outer-Nesterov the outer gradient update is applied using the Nesterov momentum update rule (Algorithm 4, lines 5 and 6).
> > >
> > > 10. For the results of Figures 3, 4, 8 and 9 we use task-specific outer hyperparameters obtained using grid searching. The optimal outer-PPO hyperparameters are found in Table 4 in Appendix C.2. In this table we observe that the optimal values can indeed vary for a given environment suite. However, the baseline PPO hyperparameters are themselves not common across an environment suite (Table 3) hence we would not expect a common set of outer hyperparameter values to be optimal across a suite. Indeed we observe that the optimal baseline PPO hyperparameters themselves vary significantly within environment suites (e.g clipping $\epsilon$ and learning rates), which will greatly affect the surrogate objective optimization and hence outer gradients. We provide the sensitivity plots in Figures 5, 6 and 7 to show how sensitive the methods are to their hyperparameters by sharing common sets of values across the grid, indicating that the methods are broadly robust on Brax and Jumanji, with reasoning provided for the high sensitivity on MinAtar in line 466 onwards.
> > >
> > > 10. Indeed several environments (ant, halfcheetah, maze) have an optimal outer learning rate $\sigma < 1$. In lines 204 - 211 we motivate this as attenuating an update we cannot fully trust due to noise in the data collection and stochastic optimization. However, the reviewer provides an insightful comment, concerning that the PPO outer gradient should be a direction towards policy improvement, otherwise the standard algorithm would not work. However, given the non-linear map from parameters to policy, and non-convex surrogate loss function this direction cannot be considered to monotonically improve performance in the range $\sigma \in [0, 1]$. We only know that performance is greater at the surrogate objective solution $\theta_k^*$ than behavior parameters $\theta_k$ but there may be higher performing parameters at points  $\theta_k + \sigma(\theta_k^* - \theta_k)$ for $\sigma \in [0,1]$. We have added this motivation to the work in lines 208 - 210.
> > >
> > > We thank you once again for dedicating your time and effort to reviewing our paper and offering insightful comments. We hope that these clarifications and amendments address your concerns, and if so, would be grateful if you could consider upgrading your score. We would of course by happy to answer any further questions you may have.
> > >
> > > [1] Deep Reinforcement Learning at the Edge of the Statistical Precipice

---

> > > > ### Comment · Reviewer_xV9r · 2024-11-30
> > > > **Thanks for the detailed reply!**
> > > >
> > > > Thanks for explaining your metric, graphs and experiment details. My main concern is not well solved so I will keep my original score.
> > > >
> > > > The main question focuses on the first algorithm, which gives dominant performances, and asks how the proposed algorithm differed from rescaling the learning rate of the original PPO?”
> > > >
> > > > It is answered as  "To help understand the distinction, we would like to draw the reviewer's attention to Algorithm 2 and Algorithm 3." My question is not directly answered and my concern is not cleared.

---

### Author Response · Authors · 2024-11-20
**General Response to Reviewers**

# General Response

We would like to thank all reviewers for the time and effort they have invested in reviewing our work, and the thoughtful comments and suggestions they have all made. We are particularly encouraged by their comments that "the idea of the proposed method (outer-PPO) is interesting", and that we "highlight the main research questions well". We are also pleased that reviewers recognise that we provide a "robust empirical analysis of the proposed algorithm (with very detailed ablation studies)", and that "the paper makes claims on its empirical performances, which are well supported by the empirical results."

We have provided a dedicated response to each reviewer, but summarize the core changes to our paper here:

### Can PPO Recover Outer-PPO?

We have added Appendix G discussing that base hyperparameters (in particular clipping $\epsilon$ and inner learning rate $\eta$) cannot recover the behavior of outer-PPO, to support our responses to reviewers xV9r and LKWg.

### Computational Complexity of Outer-PPO

We have added Appendix H discussing the computational complexity of outer-PPO, specifically that there is no material increase in time complexity. We further include Table 5 in which we show there is no increase in runtime for outer-PPO over baseline PPO, in response to reviewer z32D.

### Details on Evaluation Metrics

We have added Appendix I describing the evaluation procedure in greater detail, included the metric definitions, at the suggestion of reviewer xV9r.

### Discrepancy in Hyperparameter Tuning

We thank reviewer xVTT for highlighting a discrepancy in our hyperparameter tuning process for baseline and outer-PPO methods, in which the outer-PPO grid searches may have been more effective than the additional 100 trials of baseline tuning performed. To address this we have added added Appendix J, in which the tuning procedure of outer-PPO is the same as the baseline tuning. Preliminary results on outer-LR on Brax demonstrate improvement over baseline of comparable magnitude to the results of Figures 3, 4, 8 and 9. Further results for other algorithms and suites are to be added later in the discussion period.

## Other Changes

All changes to the paper have been made using blue text.

- Added further motivation for outer learning rates $\sigma < 1$ on lines 207 - 210.
- Removed optimality gap from Figure 3, as it is simply $1 - \text{mean}$ hence redundant.
- Added further emphasis on the common hyperparameters used in the sensitivity plots in captions of Figures 5, 6 and 7.
- Added comparison of MinAtar baseline results to those reported by other works [2], and to the task-defined maxima in lines 473.
- Added detail that we use Adam for the inner loop optimization on line 778.
- Added new table to Appendix B (Table 1 in new numbering) describing the relevant PPO implementation details as defined by [1].
- Added Appendix K, including plots of the sensitivity of standard PPO to scaling of inner learning rates and clipping $\epsilon$.
- Added red line to show cumulative maximum in baseline sweeps, and further explanation of these plots in caption.
- Moved discussion of runtime from Appendix B (Implementation Details) to new Appendix H (Computational Complexity).

We hope that our answers address your concerns, and we are looking forward to a productive discussion period.

[1] The 37 Implementation Details of Proximal Policy Optimization
[2] Discovered Policy Optimization

---

### Meta-Review · Area_Chair_Ltx1 · 2024-12-20

**Metareview:**

The paper proposes a new algorithm that creates an inner and outer loop PPO method where the inner loop estimates an update direction and the outer loop actually changes the policy. The paper performs a non-insignificant amount of hyperparameter tuning to make a performance comparison and perform some hyperparameter sensitivity analysis of the new algorithm. The primary concerns of the reviewers are around the relevance of the method and the empirical results being meaningful. To these complaints, I will add that the hyperparameter tuning while more than done in most papers, is not sufficient for accurate statistical accounting of the performance differences. Because the chosen hyperparameters are the result of a stochastic process, this uncertainty needs to be accounted for in the final results. Consequently, the statistical comparisons of the algorithms do not account for this randomness and thus, conclusions cannot be accurately drawn.

I also want to add that it is not clear what problem this new method is actually solving. It may be novel and it could work better, but it is unclear what problem with policy optimization algorithms is actually being addressed. It is also unclear how this new method actually impacts the optimization process differently than PPO, this is a common complaint made by reviewers and there should be evidence to show this effect.

**Additional Comments On Reviewer Discussion:**

There was a discussion with the reviewers, and a few increased their scores, but several had significant concerns that remained unresolved. This leads me to not recommend the paper for acceptance.

---

### Decision · Program_Chairs · 2025-01-22

Reject